# UDP-glucosyltransferase OsUGT75A promotes submergence tolerance during rice seed germination

Yongqi He[1,2], Shan Sun[1,2], Jia Zhao[1], Zhibo Huang[1], Liling Peng[1], Chengwei Huang[1], Zhengbin Tang[1], Qianqian Huang[1] & Zhoufei Wang ®[1] ✉

Submergence stress represents a major obstacle limiting the application of direct seeding in rice cultivation. Under flooding conditions, coleoptile elongation can function as an escape strategy that contributes to submergence tolerance during seed germination in rice; however, the underlying molecular bases have yet to be fully determined. Herein, we report that natural variation of rice coleoptile length subjected to submergence is determined by the glucosyltransferase encoding gene *OsUGT75A*. OsUGT75A regulates coleoptile length via decreasing free abscisic acid (ABA) and jasmonic acid (JA) levels by promoting glycosylation of these two phytohormones under submergence. Moreover, we find that OsUGT75A accelerates coleoptile length through mediating the interactions between JASMONATE ZIMDOMAIN (OsJAZ) and ABSCISIC ACID-INSENSITILE (OsABI) proteins. Last, we reveal the origin of the haplotype that contributes to coleoptile length in response to submergence and transferring this haplotype to *indica* rice can enhance coleoptile length in submergence conditions. Thus, we propose that *OsUGT75A* is a useful target in breeding of rice varieties suitable for direct seeding cultivation.

Rice (*Oryza sativa*) is one of the most important crops cultivated worldwide. Transplanting of rice seedlings into puddled fields is widely practiced in most rice growing area. Recently, direct seeding has been increasingly adopted under both rainfed and irrigated conditions, owing to multiple benefits including low cost and convenience[1,2]. To suppress weeds germination and thereby reduce the costs of manual weeding and/or the dependence on herbicides, farmers usually flood the direct seeded field, which creates submergence-related stress[3]. Unfortunately, rice is notably susceptible to submergence-related stress during the period of seed germination. Thus, expanding the application of direct seeding requires breeding rice cultivars with enhanced submergence tolerance[2–4].

To cope with submergence stress, rice employs two contrasting adaptive strategies, namely quiescence and escape[5,6]. The quiescence strategy is based on suppression of shoot elongation by *SUB1* and

promotion of survival by reducing carbohydrate consumption[5,7]. In contrast, the escape strategy is generally observed during seed germination and typically involves an increase in coleoptile and/or mesocotyl elongation to enhance survival under submergence[8,9]. Coleoptile is a cylindrical organ that unsheathe the first leaf and shoot apex in young monoct seedlings. Mesocotyl is the organ between the coleoptilar node and the basal part of the seminal root in young monoct seedlings. Under submergence condition, fast elongation of coleoptile and/or mesocotyl allows seedlings to gain access to the water surface, thereby allowing the access of $O_2$ for seed aerobic respiration under submergence, and thus meeting energy requirements for the subsequent development of leaves and roots[10]. Thus, the final length of coleoptiles and/or mesocotyl and the speed of their elongation determines the success of seedling establishment under submergence conditions. Several quantitative trait loci (QTLs) for rice

[1]The Laboratory of Seed Science and Technology, Guangdong Key Laboratory of Plant Molecular Breeding, State Key Laboratory for Conservation and Utilization of Subtropical Agro-Bioresources, South China Agricultural University, 510642 Guangzhou, China. [2]These authors contributed equally: Yongqi He, Shan Sun. ✉e-mail: wangzf@scau.edu.cn

seed submergence tolerance have been identified[11–16], but information on the genetic basis underlying these QTLs is scarce. It is reported that natural variation of *OsGSK2*, encoding a conserved GSK3-like kinase determines rice mesocotyl length by coordinating strigolactone and brassinosteroid signaling[17]. Natural variation of *calcineurin B-like protein10* (*OsCBL10*) influences coleoptile length and survival rate during rice seed germination under submergence conditions[18]. In addition, the trehalose-6-phosphate phosphatase encoding gene *OsTPP7* is involved in promoting rice anaerobic germination tolerance, reflected by increasing coleoptile length[19]. Further, a 14-3-3 protein OsGF14h acts as a signal switch to balance ABA signaling and gibberellic acid (GA) biosynthesis to enhance flooding tolerance by increasing coleoptile length during rice germination[20].

Members of the UDP-glucosyltransferases (UGTs) family typically catalyze the transfer of sugars onto small-molecules[21]. Among them, UGT75 family proteins belong to plant family 1 UDP-glycosyl-transferases, which can catalyze the glycosylation of indole-3-acetic acid (IAA), indole-3-butyric acid (IBA), and ABA in *Arabidopsis*[22–24]. The *Arabidopsis UGT75D1* and *UGT75B1* play important roles in regulating seed germination upon exposure to salt and osmotic stresses via glycosylation of IBA and ABA, respectively[23,24]. However, whether members of the UGT75 family proteins regulate coleoptile length under submergence through glycosylation of related phytohormones is unknown. It has been proposed that the degradation of ABA contributes to a rapid elongation of rice coleoptiles[25]. A recent study shows that rice OsGF14h enhances coleoptile length through regulating the interaction of ABA and GA during seed germination under flooding conditions[20]. JA has been shown as a regulatory factor associated with stem elongation in deep-water rice via GA-mediated signaling[26,27]. Moreover, the interactions between JASMONATE ZIM-DOMAIN (JAZ) proteins and ABSCISIC ACID-INSENSITIVE (ABI5) protein are involved in the suppression of seed germination in *Arabidopsis* and wheat[28]. However, ABA- and JA-dependent coleoptile elongation has not been well investigated in rice.

In the present study, we functionally characterize the UDP-glucosyltransferase encoding gene (*OsUGT75A*) under a major QTL for rice coleoptile length under submergence. We observe that OsUGT75A can glycosylate ABA and JA, which consequently reduce the levels of free ABA and JA, respectively, and thereby promote rice coleoptile growth under submerged conditions. Based on our observations, we propose that OsUGT75A-mediated crosstalk between ABA and JA signaling functions in coleoptile elongation in submerged rice. *OsUGT75A* could be useful for breeding rice varieties suitable for direct seeding.

## Results

### Genetic variation of rice coleoptile length under submergence
We screened coleoptile length of 245 rice accessions (Supplementary Data 1) selected from a core worldwide collection under submergence condition. Coleoptile length expressed a normal distribution within the range of 0–4.5 cm. Approximately 78.0% of the accessions had the coleoptile length between 1.0 and 3.0 cm (Supplementary Fig. 1). To investigate genetic control of the natural variation in coleoptile length, we conducted genome-wide association study (GWAS) using previously reported genotyping data with a minor allele frequency (MAF) ≥ 5%[29,30]. The results revealed a major locus (*qCL11*) on chromosome 11 (Fig. 1a, b). Interestingly, our additional GWAS analyses using subset of *japonica* and *indica* accessions showed that *qCL11* could only be detected in the *japonica* population (Supplementary Fig. 2). Thus, it's likely that genetic variation of rice coleoptile length in response to submergence may be exclusive to *japonica* genotypes. To look for candidate causal genes underlying *qCL11*, we checked the genomic region covering 100-kb upstream and downstream of the leading SNP of *qCL11*. According to the assembled genome, there are nine annotated genes within this region (Supplementary Table 1), with

the leading SNP containing a nonsynonymous nucleotide substitution located in the coding region of *LOC_Os11g25990* (Fig. 1c).

By analyzing the publicly available gene expression data (https://www.genevestigator.com), we found that the expression of *LOC_Os11g25990* is most significantly induced during seed germination in *japonica* Nipponbare under anaerobic conditions (Fig. 1d), which indicates that *LOC_Os11g25990* might be the candidate causal gene. We subsequently analyzed SNPs occurring in the coding sequence and 2-kb upstream and 1-kb downstream of the gene (Supplementary Data 2). We identified two haplotypes of *LOC_Os11g25990* (Fig. 1e). Haplotype 1 (Hap1) (functional SNP1 "G" at position 14850366 in the promoter region; functional SNP2 "A" at position 14849038 in the coding region) was found to be associated with longer coleoptiles, mainly among *japonica* accessions, compared with haplotype 2 (Hap2), which tended to be the characteristics of the accessions with shorter coleoptiles (Fig. 1f, g). Furthermore, we used a near isogenic line (NIL) derived from a cross between *indica* Kasalath (short coleoptile length; Hap2) and *japonica* Koshihikari (long coleoptile length; Hap1) to validate the function of *qCL11* (Fig. 1h). When subjected to submergence, NIL plants harboring the transferred Hap2 (Kasalath) in the Koshihikari background exhibited shorter coleoptiles compared with that of Koshihikari (Fig. 1i, j). Collectively, these evidences support that *LOC_Os11g25990* might be the causal gene for *qCL11*.

### OsUGT75A regulates coleoptile length under submergence
The Arabidopsis ortholog of *LOC_Os11g25990* has been annotated as a UDP-glucosyltransferase (Supplementary Fig. 3 and Supplementary Data 3). Based on phylogenetic grouping, we named the candidate gene as *OsUGT75A*. To further confirm that *OsUGT75A* is the causal gene of *qCL11*, we constructed three knock-out lines (*Osugt75a-1*, *Osugt75a-2*, and *Osugt75a-3*) in *japonica* Nipponbare background using the clustered regularly interspaced short palindromic repeats (CRISPR)/CRISPR-associated protein 9 (Cas9) system (Supplementary Fig. 4) and generated two *OsUGT75A* overexpressing lines (OE-1 and OE-2). The seeds of homozygous lines were germinated under water of different depths (Fig. 2 and Supplementary Fig. 5). The results revealed that *Osugt75a* plants had shorter coleoptile length than that of Nipponbare plants (<1 cm vs. 3–4 cm; Fig. 2a, b). In contrast, the coleoptile length and seedling establishment of the OE-1 and OE-2 lines were significantly higher than that of the control plants (Fig. 2c–e).

To further characterize the role of *OsUGT75A* in regulating of coleoptile length, we conducted qRT-PCR to investigate the expression patterns of *OsUGT75A* during seed germination under submergence. Consistent with the aforementioned results, we observed that the expression of *OsUGT75A* was markedly enhanced under submergence compared with that under normal conditions (Fig. 2f). In addition, we analyzed the expression pattern of *OsUGT75A* using transgenic plants carrying a *β-glucuronidase* (*GUS*) reporter gene. Consistent with the qRT-PCR results, we detected strong GUS staining in developing coleoptiles under submergence (Fig. 2g). These observations thus confirm that *OsUGT75A* is the causal gene of *qCL11*.

### OsUGT75A regulates coleoptile length via glycosylation-mediated fine-tuning of free ABA and JA levels
Previous reports showed that UGT75 family proteins can catalyze the glycosylation of IAA, IBA, and ABA in plants[22–24]. We firstly compared the levels of hormones in germinating seeds of WT and *Osugt75a* mutants after 3 days submergence. Our metabolomics data revealed that the levels of free ABA, JA, and JA-Ile were significantly higher in mutant plants (Supplementary Fig. 6a–c), whereas the level of gibberellin ($GA_{24}$) was lower in *Osugt75a* mutants than that in WT plants (Supplementary Fig. 6e). In contrast, we didn't detect significant differences between WT and mutant plants with respect to the levels of the ethylene precursor 1-aminocyclopropane-1-carboxylic acid (ACC),

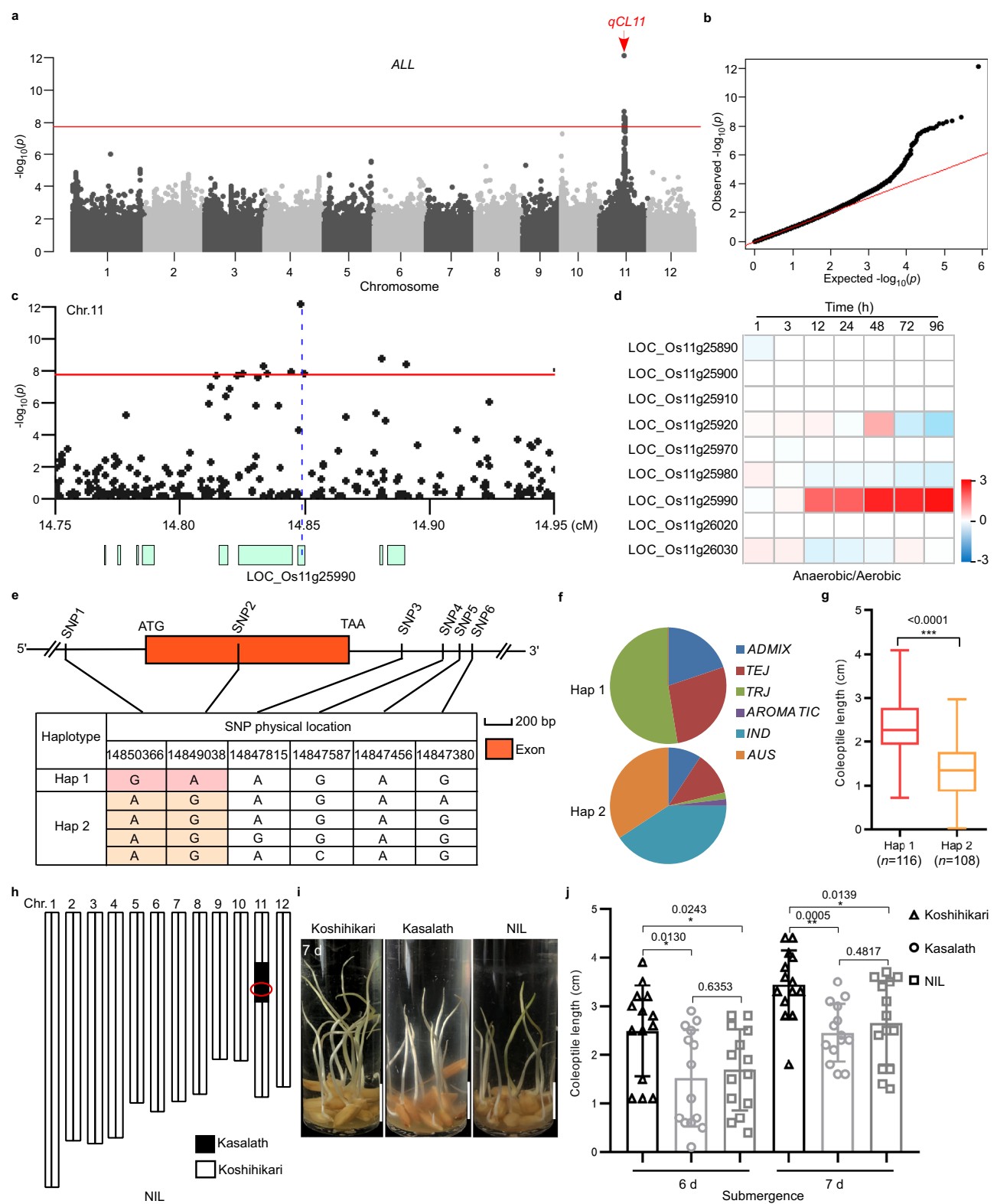

auxins (ICA, IAA, and IPA), or cytokinins (cZ, IP, and tZ) (Supplementary Fig. 6f–l). On the basis of these observations, we speculated that the shorter coleoptiles of *Osugt75a* mutants might be attributed to the upregulated accumulations of ABA and JA and decreased accumulation of GA. To further confirm this conjecture, we examined the effects of exogenous application of ABA, JA, GA₃, IAA, and ethylene analog ethephon (ETH) on the growth of *Osugt75a* mutant and WT plant coleoptiles. We observed that the length of *Osugt75a* mutant

coleoptiles was significantly reduced by ABA or JA treatments (Fig. 3a–d), though no appreciable responses to GA₃, ETH, or IAA treatments were observed (Supplementary Figs. 7 and 8). Interestingly, the GA₃ treatment increased the coleoptile elongation of WT but not in *Osugt75a* mutants (Supplementary Fig. 7). Further, our qRT-PCR analysis showed that the expression of *OsUGT75A* was significantly induced by GA₃ treatments in WT during seed germination under submergence (Supplementary Fig. 9). It suggests that the GA

**Fig. 1 | Identification on rice chromosome 11 of a QTL, *qCL11*, for coleoptile length under submergence. a** Manhattan plots for the whole population of rice accessions. The red arrow indicates the identified locus. **b** A Quantile–Quantile plot. **c** Identification of candidate genes in the *qCL11* region. In **a**–**c** no data adjustments were made for GWAS with a threshold of $2.50 \times 10^{-8}$ (0.01 significance level). **d** *LOC_Os11g25990*, the putative candidate gene for *qCL11*, was significantly induced during seed germination under anaerobic conditions using publicly available microarray data (http://www.genevestigator.com). Red, up-regulation; Blue, down-regulation. Values represent the $\log_2$-fold gene changes. **e, f** Haplotypes of *LOC_Os11g25990* identified in the coding sequence (CDS), 2-kb region upstream and 1-kb region downstream of the gene. *IND indica, TEJ* temperate *japonica, TRJ* tropical *japonica.* **g** Box-plots of coleoptile length in accessions containing the

different haplotypes. *n* = 116/108 accessions. Center lines show the medians, box limits indicate the 25th and 75th percentiles, whiskers extend to the minimal and maximal values as determined by GraphPad Prism 8.0.1 software. **h** Chromosomal location of the introgressed Kasalath (with the Hap2 allele) segments in a near isogenic line (NIL) in a Koshihikari (with the Hap1 allele) background. **i** Representative images of Koshihikari (with the Hap1 allele) and NIL (with the Hap2 allele) under submergence (8 cm depth of water) for 7 days. Scale bars represent 10 mm. **j** Coleoptile length under submergence (8 cm depth of water) for 6 and 7 days. Data were presented as mean ± SD, *n* = 14. In **g, j** significant differences were determined by two-tailed Student's *t* tests (\**P* < 0.05, \*\**P* < 0.01, \*\*\**P* < 0.001). Source data are provided as a Source Data file.

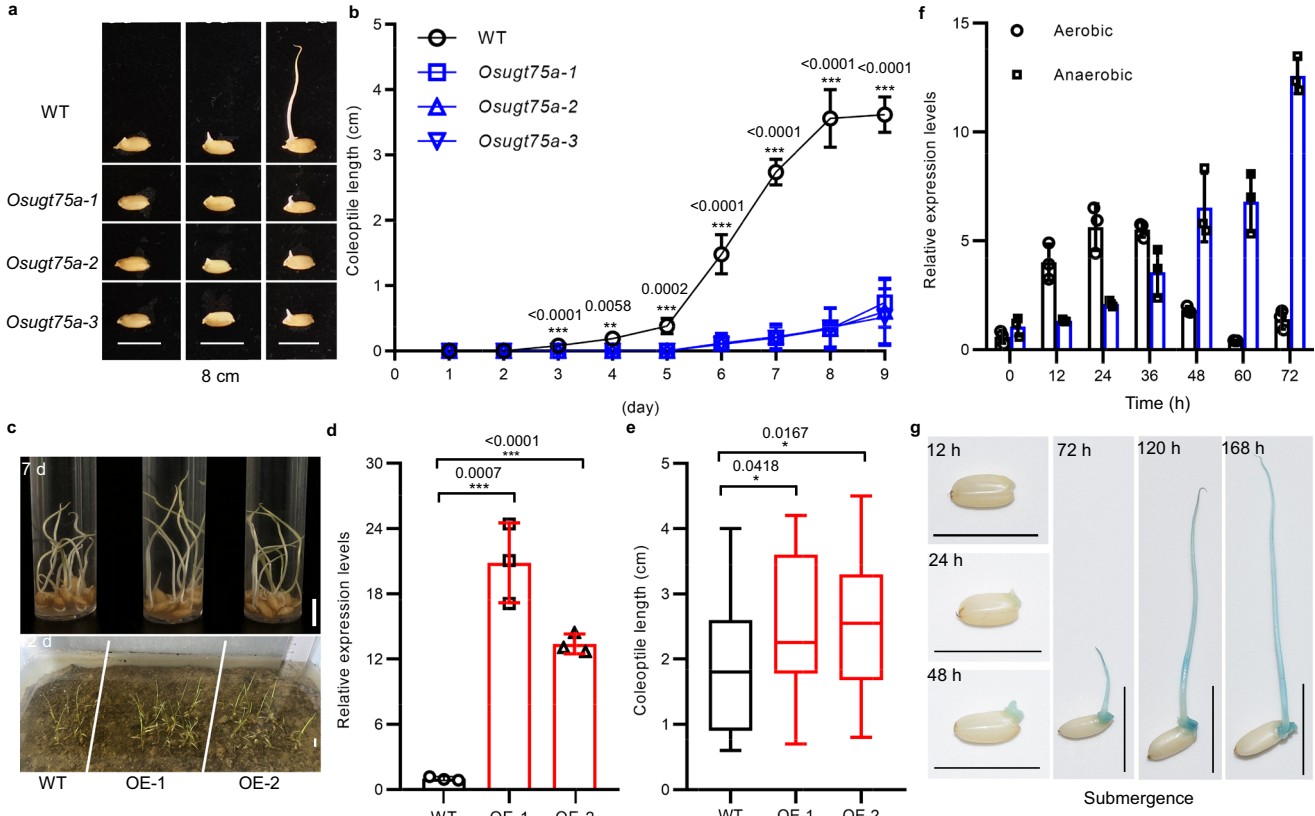

**Fig. 2 | Characterization of the regulation of rice coleoptile length under submergence by *OsUGT75A*. a** Representative images of coleoptile length and **b** dynamic changes in coleoptile length among Nipponbare wild-type (WT) and *Osugt75a* mutant under submergence (8 cm depth of water). Data were presented as mean ± SD, *n* = 5. **c** Representative images of coleoptile length under submergence (8 cm depth of water) for 7 days and seedling establishment of direct seeding in soils under submergence (4 cm depth of water) for 12 days among WT and overexpressed *OsUGT75A* lines (OE-1 and OE-2). **d** Comparison of the relative expression levels of *OsUGT75A* in WT and overexpressed lines determined by qRT-PCR analysis. Data were presented as mean ± SD, *n* = 3 biologically independent samples. Expression is shown relative to that in the WT, the value of which was set to 1, with the *OsActin* gene being used as an internal control. **e** Comparison of the

coleoptile lengths of WT and overexpressed lines. *n* = 23/22/26. Center lines show the medians, box limits indicate the 25th and 75th percentiles, whiskers extend to the minimal and maximal values as determined by GraphPad Prism 8.0.1 software. **f** The expression patterns of *OsUGT75A* in germinating seeds under normal and submerged conditions determined by qRT-PCR analysis. Data were presented as mean ± SD, *n* = 3 biologically independent samples. Expression is shown relative to that at 0 h under non-submerged conditions, the value of which was set to 1, with the *OsActin* gene being used as an internal control. **g** Histochemical staining for β-glucuronidase (GUS) activity in germinating seeds. Scale bars represent 10 mm. In **b, d, e** significant differences were determined by two-tailed Student's *t* tests (\**P* < 0.05, \*\*\**P* < 0.001). Source data are provided as a Source Data file.

promotion of coleoptile growth might be through increasing *OsUGT75A* expression. Moreover, we showed that *Osugt75a* plants were more ABA- and JA-sensitive than WT plants, and significantly lower coleoptile length were observed under ABA + JA treatment compared with those of individual ABA or JA treatment (Fig. 3a–d). Meanwhile, the further dynamic analysis confirmed that the levels of free ABA, JA, and JA-Ile were significantly higher in the germinating seeds (Fig. 3e–g) and in the developing coleoptiles (Fig. 3h–j) in *Osugt75a* mutants compared to those of WT under submergence.

These observations thus indicate that the knockout of *OsUGT75A* results in diminished coleoptile length, which is assumed to be mediated via ABA and JA signaling pathways.

As mention above, the free ABA and JA was significantly enhanced in *Osugt75a* plants. Next, fusion proteins of OsUGT75A with the GST tag were heterologously expressed in *Escherichia coli* and then quantified for the enzyme reaction (Fig. 4a). High-performance liquid chromatography (HPLC) analysis of the enzymatic reaction products confirmed that OsUGT75A could catalyze the glycosylation of ABA.

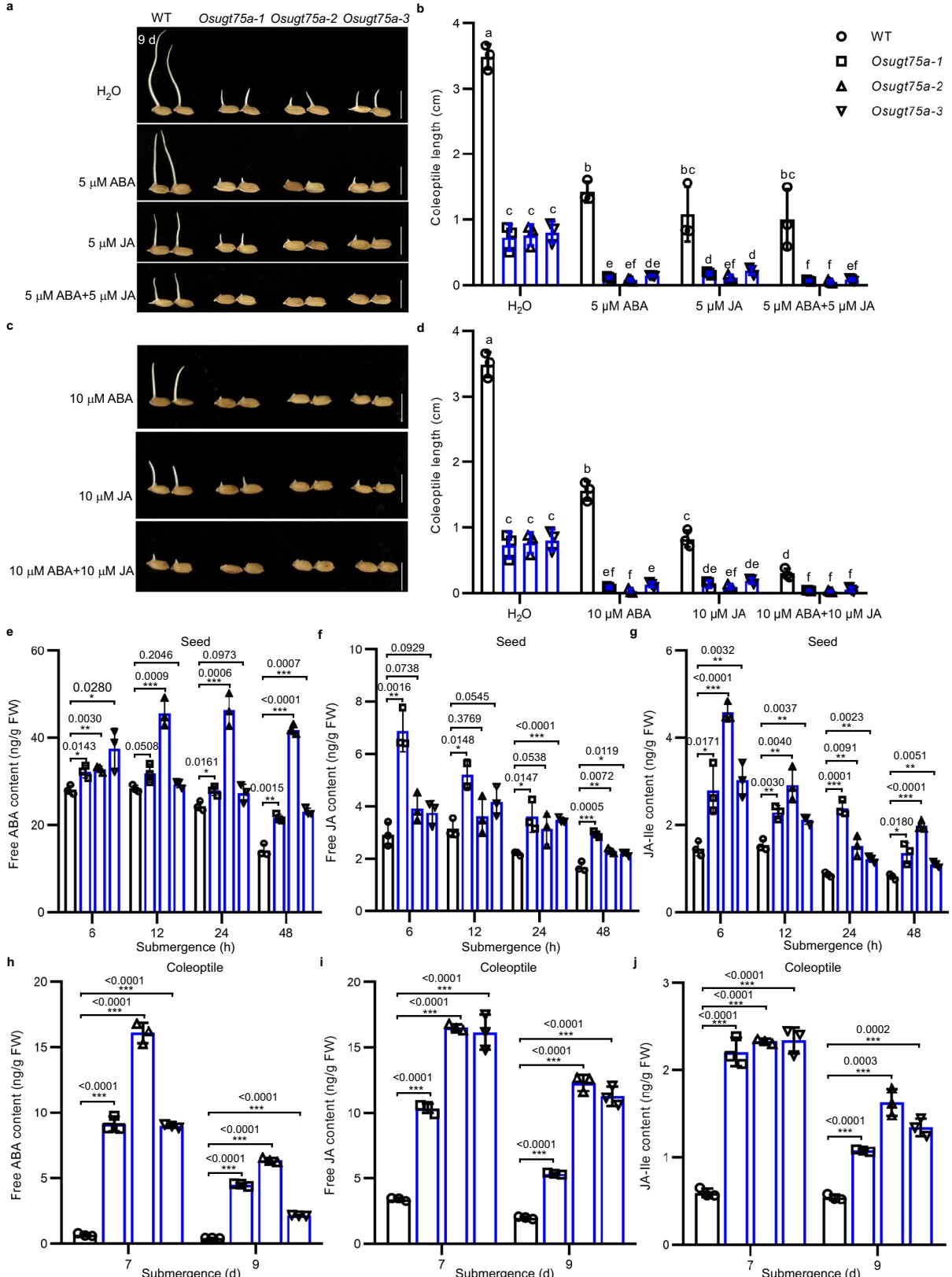

The ABA-glucose conjugate (ABA-Glu) with exactly the same retention time as the authentic standard was detected among the enzymatic reaction products (Fig. 4b). Meanwhile, HPLC analysis showed that OsUGT75A could potentially catalyze the glycosylation of JA reactions (Fig. 4c). Since JA-glucose conjugate (JA-Glu) is commercially unavailable, mass spectrometry (MS) analysis was used to further verify the putative JA-Glu formed in the reaction (Fig. 4d). The molecular weight (M) of JA-Glu is 372.27. MS analysis in the positive mode showed that the ion peaks of the reaction product were at $m/z$ 395.2202 (M + Na$^+$), $m/z$ 210.1180 (M-H-Glu), and $m/z$ 166.0911 (M-COOH-Glu), which is consistent with the expected protonated molecular ions of JA-Glu (Fig. 4d), suggesting the putative JA conjugate is the JA-Glu.

**Fig. 3 | OsUGT75A regulates rice coleoptile length under submergence (8 cm depth of water) by influencing abscisic acid (ABA) and jasmonic acid (JA) levels. a, c** Representative images of the coleoptile length of WT and *Osugt75a* mutants in response to 5 and 10 μM ABA and/or JA treatments after 9 days of submergence. Scale bars represent 10 mm. **b, d** A comparison of coleoptile lengths in WT and *Osugt75a* mutants in response to control (H₂O), ABA, and JA treatments after 9 days of submergence. The contents of free ABA, JA, and JA-Ile in **e**–**g** the germinating seeds and in **h**–**j** the developing coleoptile in Nipponbare wild-type (WT) rice and *Osugt75a* mutants under submergence. In **b, d** data were presented as mean ± SD, *n* = 3 biologically independent experiments; different letters indicate significant differences (*P* = 0.05, one-way ANOVA). In **e**–**j** data were presented as mean ± SD, *n* = 3 biologically independent samples; significant differences were determined by two-tailed Student's *t*-tests (*\**P* < 0.05, *\*\**P* < 0.01, *\*\*\**P* < 0.001). Source data are provided as a Source Data file.

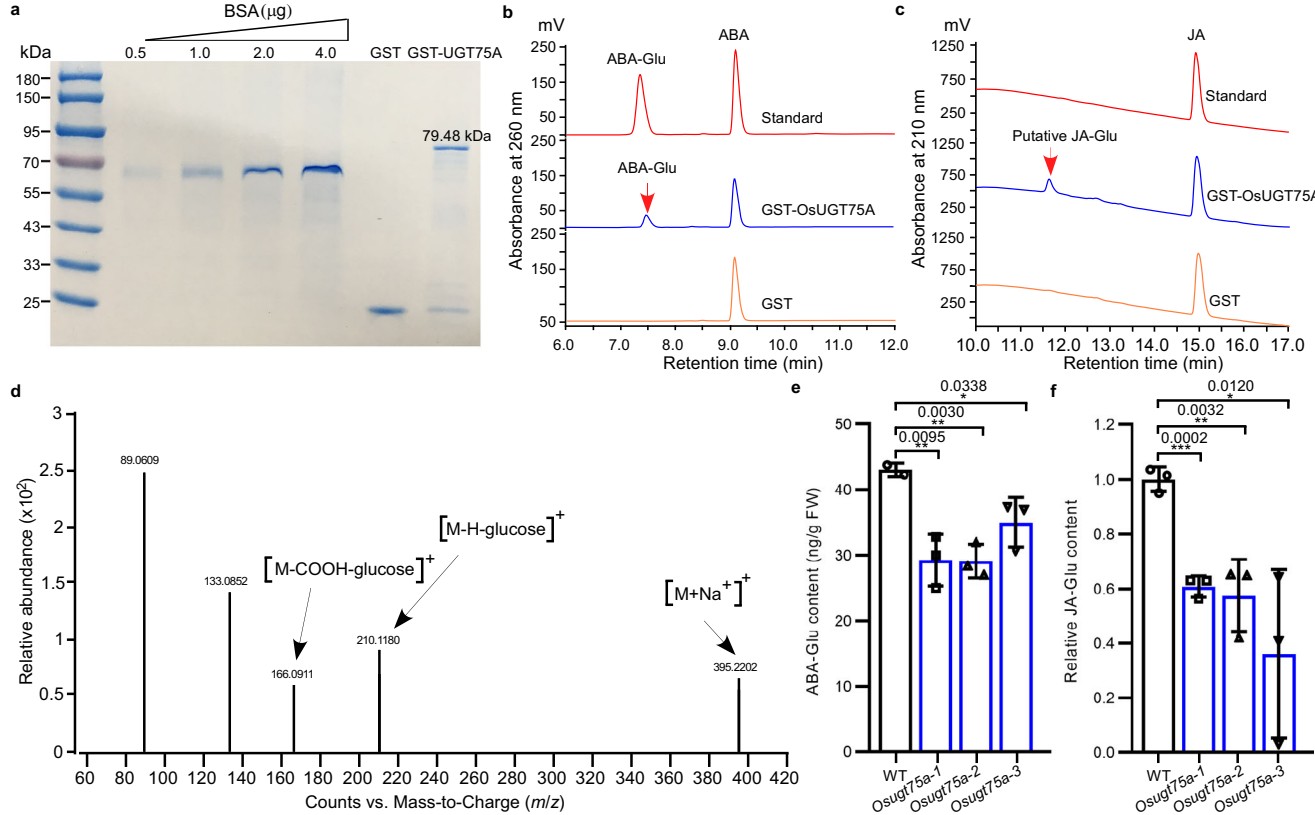

**Fig. 4 | Glycosylation ABA and JA by OsUGT75A. a** The full-length GST-OsUGT75A fusion protein was quantified using bovine serum albumin protein standard (BSA). *n* = 3 independent experiments. The full-length GST-UGT75A fusion protein catalyzes the glycosylation of **b** ABA and **c** JA in vitro, but the GST tag could not. Arrow indicates the product ABA-Glu and putative JA-Glu. **d** JA-Glu formed in **c** was confirmed by MS/MS analysis in positive ionization mode. The molecular weight (M) of JA-Glu is 372.27. The reaction products had the ion peaks at *m/z* 395.2202 (M + Na⁺), *m/z* 210.1180 (M-H-Glu), and *m/z* 166.0911 (M-COOH-Glu). Quantification of absolute amounts of **e** the ABA-Glu and **f** the relative levels of JA-Glu in coleoptiles in Nipponbare wild-type (WT) and *Osugt75a* mutants under submergence (8 cm depth of water) for 7 days. Absolute amounts of ABA-Glu were measured by HPLC using the commercial available chemical standard. Relative levels of JA-Glu were determined by targeted HPLC. The level of JA-Glu is shown relative to that in WT, the value of which was set to 1. In **e, f** data were presented as mean ± SD, *n* = 3 biologically independent samples; significant differences were determined by two-tailed Student's *t* tests (*\**P* < 0.05, *\*\**P* < 0.01, *\*\*\**P* < 0.001). Source data are provided as a Source Data file.

Furthermore, we observed that the contents of ABA-Glu and the relative levels of JA-Glu were significant lower in coleoptiles of *Osugt75a* mutants than those in Nipponbare wild-type (WT) under submergence (Fig. 4e, f and Supplementary Fig. 10). Moreover, IAA-O-glucoside was not detected (Supplementary Fig. 11). These data suggest that OsUGT75A reduced the level of free ABA and JA by catalyzing the glycosylation ABA and JA during submergence.

The submergence-induced ABA 8′-hydroxylase (*OsABA8ox*) gene encodes the enzyme that catalyzes the oxidation of ABA, which results in the degradation of ABA for submergence tolerance in rice[31,32]. We observed that the expressions of *OsABA8ox* genes, especially *OsABA8ox1* in the early (0–12 h) stage of seed germination and *OsABA8ox3* in the middle (12–48 h) stage of seed germination, were significantly decreased in *Osugt75a* mutants compared to those of WT under submergence (Supplementary Fig. 12). It suggests that *OsABA8ox* genes might be also involved in the *OsUGT75A*-mediated degradation of

ABA under submergence. As mentioned above, the expression of *OsUGT75A* is gradually increased especially in the late (48–72 h) stage of seed germination (Fig. 2f) and in the developing coleoptiles under submergence (Fig. 2g). These results imply that the slow induction of *OsUGT75A* may play a role in maintaining the low free ABA, JA, and JA-Ile levels in the germinating seeds and in the developing coleoptiles under submergence. In summary, both in vitro and *in planta* evidences support that OsUGT75A regulates coleoptile length in rice subjected to submergence via glycosylation-mediated fine-tuning of ABA and JA levels.

**Crosstalk between ABA and JA signaling involves in coleoptile length**

To confirm the putative roles of ABA and JA signaling in *OsUGT75A*-regulated coleoptile length, we compared the expression of *OsJAZ* and *OsABI* genes, important components in the JA and ABA signaling

pathway, respectively, in the germinating seeds of *japonica* Nipponbare under anaerobic condition using publicly available data (https://www.genevestigator.com). We found that the majority of *OsJAZs*, particularly *OsJAZ1/2/3/6/7/8/10/12*, were significantly induced in germinating seeds compared with those in dry seeds. However, the expression of *OsABI3/5* was significantly down-regulated (Supplementary Fig. 13a). To confirm these results, we conducted qRT-PCR analysis to compare the expression of *OsJAZs* and *OsABIs* between WT Nipponbare and *Osugt75a* mutants. The results revealed significant lower expression of *OsJAZ1/2/6/7/8* in the *Osugt75a* mutants than that in WT plants after 2 days of submergence (Supplementary Fig. 13b–i). In contrast, the *Osugt75a* mutants had significantly higher expression of *OsABI3/5* comparing to that in WT plants (Supplementary Fig. 13j, k). We therefore speculate that the shorter coleoptiles of *Osugt75a* mutants under submergence could be attributed to the upregulated expression of *OsABI3/5* and down-regulated expression of *OsJAZs*.

The lengths of coleoptiles under submergence were significantly reduced in *Osugt75a* mutants and WT plants in response to the combined ABA and JA treatments, compared with independent ABA or JA treatment. We thus hypothesized that the *OsUGT75A*-mediated regulation of coleoptile length under submergence might involve crosstalk between the ABA and JA signaling pathways. To confirm this, we performed a yeast two-hybrid (Y2H) assay using OsJAZs as bait and OsABI3/5 as prey. The results revealed that OsABI3/5 interacted with OsJAZ6/7 (Fig. 5a–d). This interaction was also verified by an in vitro pull-down assay, in which glutathione-*S*-transferase (GST)-fused OsABI3/5 was found to bind MBP-OsJAZ6/7, whereas GST alone did not (Fig. 5e–h). This interaction was further confirmed in vivo using bimolecular fluorescence complementation (BiFC) assays in rice protoplasts, in which the OsABI3/5 and OsJAZ6/7 proteins were fused to the N- and C-terminal regions of the yellow fluorescence protein (YFP), respectively. We detected strong YFP fluorescence signals in protoplast nuclei when OsABI3/5 and OsJAZ6/7 fusions were examined together (Fig. 5i). Taken together, these data indicated that OsABI3/5 interacts with OsJAZ6/7 both in vitro and in vivo.

We subsequently investigated whether OsJAZ6/7 affects the transcriptional activity of OsABI3/5 based on a dual-luciferase (LUC) reporter approach conducted using rice protoplasts, in which the LUC gene was fused to the promoter of *EARLY METHIONINE-LABELED 1* (*OsEM1*), a direct target of *OsABIs*[33]. The effector constructs contained *OsABI3/5* and *OsJAZ6/7* driven by the CaMV 35 S promoter (Fig. 6a). As expected, OsABI3/5 induced a marked increase in the expression levels of LUC driven by the *OsEM1* promoter, compared with expression in protoplast harboring the empty pGREENII62-SK vector. In contrast, cotransfection of 35 S::OsJAZ6/7 and 35S::OsABI3/5 disrupted the OsABI3/5-activated LUC expression (Fig. 6b, c), thereby indicating that OsJAZ6/7 play negative regulatory roles in the transcriptional activity of OsABI3/5. Moreover, we constructed knock-out lines of *OsJAZ6/7* and *OsABI3/5* in a *japonica* Nipponbare or Zhonghua11 backgrounds using the CRISPR/Cas9 system (Supplementary Fig. 14). The seeds of homozygous lines were germinated under submergence conditions. The results revealed that *Osjaz6/7* plants had shorter coleoptile length than that of the WT plants, while the coleoptile lengths of the *Osabi3* lines were significantly longer than those of WT plants (Fig. 6d, e). The phenotype of *Osabi5* mutant was not tested due to its significantly reduction of seed vigor after one month storage (Supplementary Fig. 15). Overall, *OsUGT75A*-regulated coleoptile length involves in the crosstalk between ABA and JA signaling.

## An elite haplotype of *OsUGT75A* contributes to submergence tolerance during seed germination in rice

To further investigate the evolutionary history of *OsUGT75A*, we constructed a haplotype network using all *OsUGT75A* haplotypes of 2058 accessions comprising 446 Asian wild rice accessions and 1612 Asian cultivated rice accessions (https://venyao.xyz/ECOGEMS/)[34]. Among these, we established that the majority of haplotypes in *japonica* (*TEJ*, temperate *japonica*; *TRJ*, tropical *japonica*) accessions are originated from of *Oryza rufipogon* III (Or-III), whereas the majority haplotypes in *indica* (*IND* and *AUS*) accessions are originated from a different group of *O. rufipogon* (Or-II) (Fig. 7a). The diversity of haplotype 1 (Hap1) was further investigated in rice accessions. Interestingly, Hap1 was mainly observed in the *japonica* accessions. Nearly all *TRJ* accessions and half of the *TEJ* accessions were found to contain elite Hap1, whereas virtually all *indica* accessions were found to lack this haplotype (Fig. 7b).

To further reveal the breeding selection of *OsUGT75A*, we investigated the sequences of *OsUGT75A* in *japonica* and *indica* varieties that are currently cultivated in China (Supplementary Data 4). The results showed that the majority of *japonica* varieties contained the elite Hap1. However, none of the tested *indica* varieties harbor this haplotype (Fig. 7c), which indicates that the elite Hap1 was not obviously selected during *indica* rice breeding. Based on these results, we developed a simple sequence repeat (SSR) marker for *OsUGT75A* that can be used to identify elite rice accessions with submergence tolerance for future breeding (Fig. 7d). Moreover, overexpression of *OsUGT75A* in *indica* variety Huanghuazhan increased its submergence tolerance during germination (Fig. 7e–g). The increased number of secondary branch and number of grains per panicle were observed in overexpression *OsUGT75A* lines compared to those of Nipponbare WT, while the plant height, grain length, grain width, and grain weight were decreased in overexpression lines (Supplementary Fig. 16). These findings imply that *OsUGT75A* could serve as a potential target to breed varieties with enhanced submergence tolerance, especially in *indica* rice.

## Discussion

Length and elongation rate of coleoptile and/or mesocotyl are the crucial developmental traits deterring the success of direct seeding in many cereal crops. Mesocotyl elongation is responsive to abiotic stresses, such as deep sowing drought, submergence, chilling, and salinity[9]. In the case of rice submerged seed germination, the long and fast elongated coleoptiles can promote submergence tolerance by providing $O_2$ when they access to the water surface[10]. Knowledge on the genetic factors and molecular mechanisms of coleoptile development under submergence condition remains limited. Here, we identified that natural variation of *OsUGT75A* plays a key role in determining rice coleoptile length under submergence. OsUGT75A could catalyze the glycosylation of the ABA and JA participated in the coleoptile elongation via integrating the ABA and JA signaling. Thus, our results contribute to the understanding of the escape strategy of submergence tolerance in rice, which might be helpful to improve submergence tolerance in other cereal crops.

Members of *Arabidopsis* UGT75 typically mediate the glycosylation ABA and IAA hormones[22-24]. In this study, whereas OsUGT75A catalyzes the glycosylation of the ABA and JA, it does not appear to play a similar role with respect to IAA. The reductions of free ABA and JA in WT plants might be attributable to the OsUGT75A glycosylation of ABA and JA. Moreover, we observed that *Osugt75a* mutants may enhance ABA by suppressing ABA oxidation. However, we do not know how OsUGT75A regulates ABA oxidation, which deserves exploration in our future study. It has been reported that GA, ethylene, and IAA enhance coleoptile growth under submergence conditions, whereas ABA and cytokinins have inhibitory effects in this regard[35]. In this study, we observed that GA enhanced coleoptile growth under submergence might be partly through increasing *OsUGT75A* expression. However, our data showed that the *OsUGT75A* regulation of coleoptile growth plausibly involves the ABA and/or JA signaling pathways while not by $GA_3$, IAA or ethylene. JA has been shown to be a regulatory factor associated with stem elongation in deep-water rice via GA-mediated signaling[26,27]. Recently, JA has been found to play critical signaling-related roles in seed germination via ABA-mediated signaling[28]. An

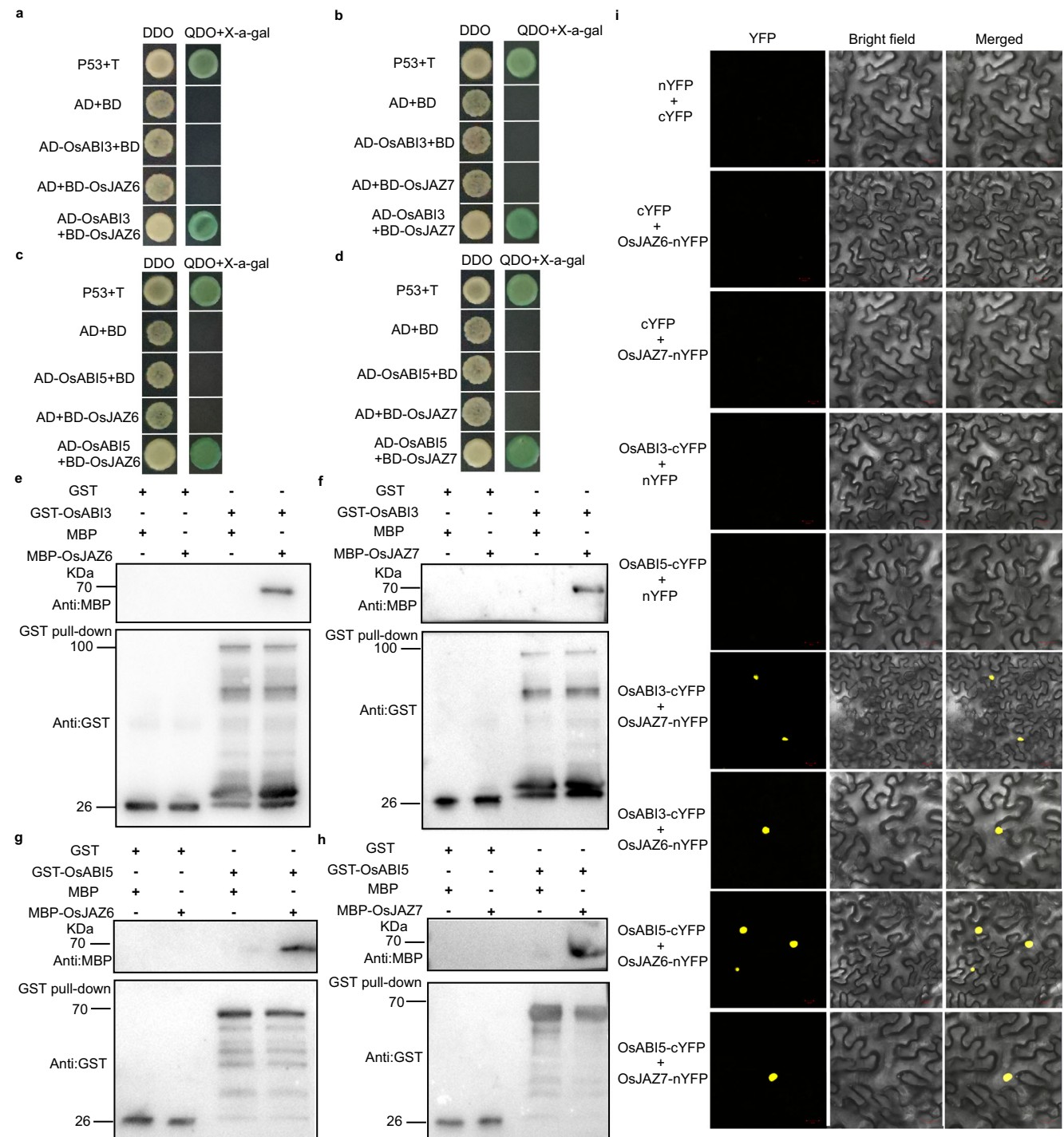

**Fig. 5 | In vitro and in vivo interaction assays between OsJAZs and OsABIs.**
**a–d** Yeast-two-hybrid (Y2H) analysis revealed interactions between OsABI3/5 and OsJAZ6/7. Transformed yeast cells were grown on SD-Trp/-His/-Leu/-Ade medium. **e–h** In vitro pull-down assays revealed interactions between OsABI3/5 and OsJAZ6/7. *n* = 3 independent experiments. **i** Bimolecular fluorescence complementation analysis of the interactions between OsABI3/5 and OsJAZ6/7. *n* = 3 biologically independent experiments. Fluorescence, emitted as a consequence of complementation of the N-terminal region of the yellow fluorescence protein (YFP) fused to OsABI3/5 (OsABI3/5-nYFP) with the C-terminal region of YFP fused to OsJAZ6/7 (OsJAZ6/7-cYFP), was observed in the nuclei of rice protoplasts. No signal was observed in the negative controls. Bars represent 50 mm. Source data are provided as a Source Data file.

accumulation of JA results in the degradation of JAZ proteins via ubiquitin-dependent SCFCOI1 E3 ligase activity, which releases ABI3/ ABI5 in response to a reduction in the interaction between JAZ proteins and ABI3/ABI5[36,37]. We sought to establish whether JA functions as an important regulatory factor for rice coleoptile length under submergence via ABA-mediated signaling. Consistent with the findings of previous studies, we found that OsJAZ6/7 proteins physically interact with OsABI3/5 and OsJAZs repress the transcriptional activity of OsABI3/5. We showed that *OsUGT75A* accelerates coleoptile length by enhancing *OsJAZ6/7* expression and then reducing *OsABI3/5* expression, suggesting it regulates submergence tolerance involving the crosstalk between ABA and JA pathways.

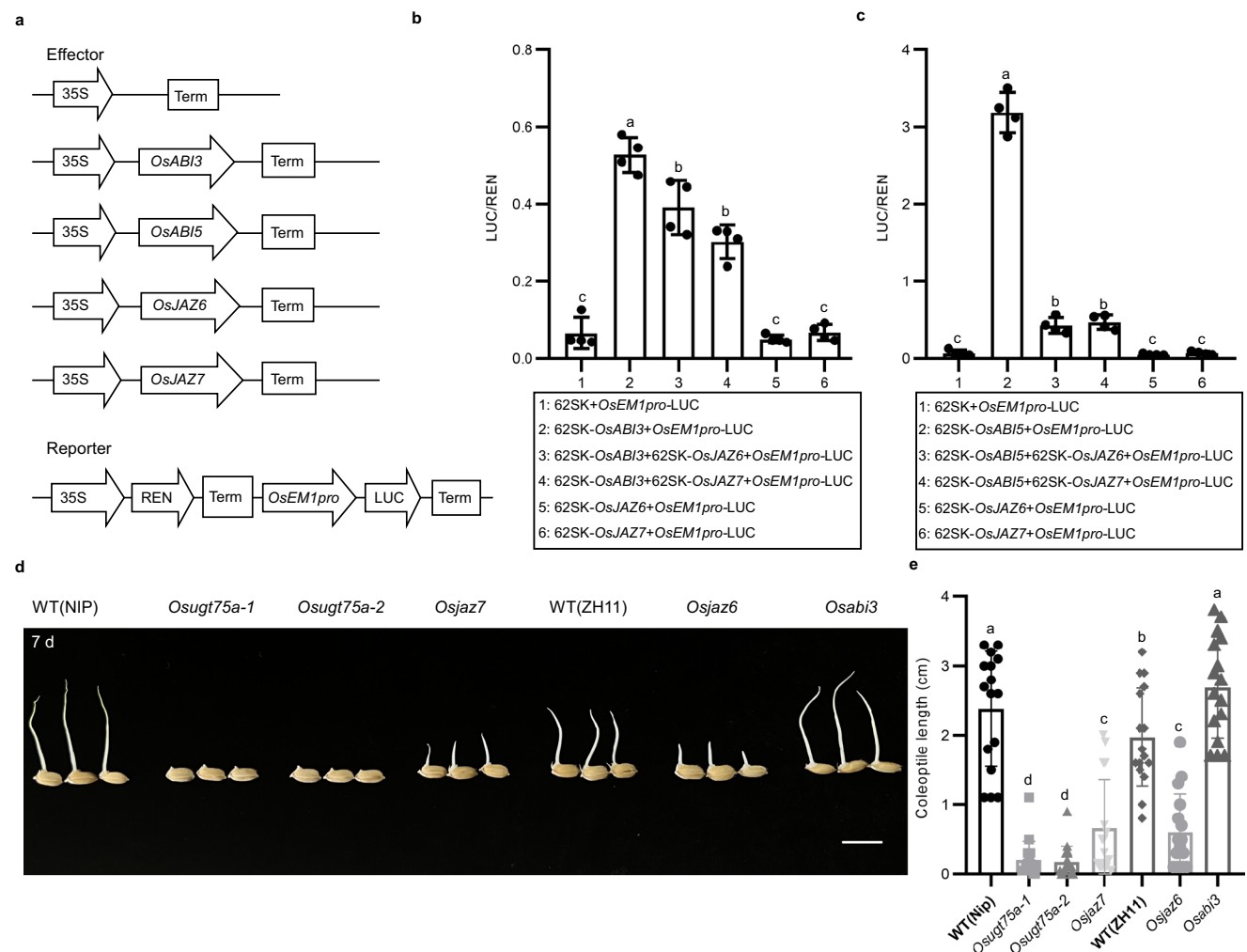

**Fig. 6 | OsUGT75A regulates coleoptile length involving the ABA and JA signaling pathways. a** Schematic diagrams of the reporters and effectors used in transient transactivation assays. **b**, **c** Transient dual-luciferase (LUC) reporter assays revealed that the OsABI3/5 activation of ABA response gene (*OsEM1*) expression could be repressed by OsJAZ6/7. Data were presented as mean ± SD, *n* = 4 biologically independent experiments. **d** Representative images of the coleoptile length of Nipponbare (NIP) or Zhonghua11 (ZH11) wild-type (WT), *Osjaz6/7*, and *Osabi3* mutants under submergence (8 cm depth of water) for 7 days. Scale bars represent 10 mm. **e** A comparison of coleoptile lengths in WT, *Osjaz6/7*, and *Osabi3* mutants under submergence. Data were presented as mean ± SD, *n* = 16. In **b**–**e** different letters indicate significant differences (*P* = 0.05, one-way ANOVA). Source data are provided as a Source Data file.

The findings of previous studies have also revealed that the coleoptiles of *japonica* varieties tend to elongate more rapidly than those of *indica* varieties[15]. Consistently, we found that when subjected to submergence, the average coleoptile length of *japonica* varieties was longer than those of *indica* varieties. Moreover, we observed that the lengths of *TEJ* accession coleoptiles were shorter than those of *TRJ* accessions. Collectively, these findings indicate that the germinating seeds of *japonica* accessions are better adapted to tolerate the anoxic conditions associated with submergence[15], although details of the underlying genetic control are yet to be fully elucidated. Previously, it has been proposed that differential flooding tolerance in rice is linked to the natural variation in the *OsCBL10* promoter regions that affect *OsCBL10* expression, and that flooding-tolerant type promoters are only present in lowland *japonica* cultivars[18]. In the present study, we confirmed that natural variation of *OsUGT75A* is associated with the regulation of coleoptile length. Interestingly, the findings of our population genetic analyses indicate that elite Hap1 originates from *Or*-III, the progenitor of temperate *japonica* varieties[38], and the majority of the *TRJ* and approximately half of the *TEJ* accessions contain this haplotype.

Moreover, we identified the elite Hap1 of *OsUGT75A* as a potential target for breeding varieties with seed germination submergence tolerance using a marker-assisted selection and transgenic approaches, which could make a significant contribution to the direct seeding cultivation of rice.

In conclusion, the findings of this study provide insights into the mechanisms underlying the submergence tolerance during seed germination in rice. OsUGT75A catalyzes the glycosylation of ABA and JA to decrease the free ABA and JA accumulation, which contributes OsJAZ6/7 proteins interacting directly with OsABI3/5 and repressing its transcriptional activity. The crosstalk of ABA and JA pathways is involved in the coleoptile elongation during seed germination to enhance submergence tolerance (Fig. 8). Gaining a thorough understanding of the mechanisms associated with submergence tolerance is not only of fundamental scientific significance but also important from the perspective of breeding rice and other cereals with elevated levels of submergence tolerance. Some progress has already been made in the breeding of advanced rice lines conferring submergence tolerance during seed germination[2]. The natural variations in *OsUGT75A* suggest that it as a potential target for breeding varieties to improve submergence tolerance in the future.

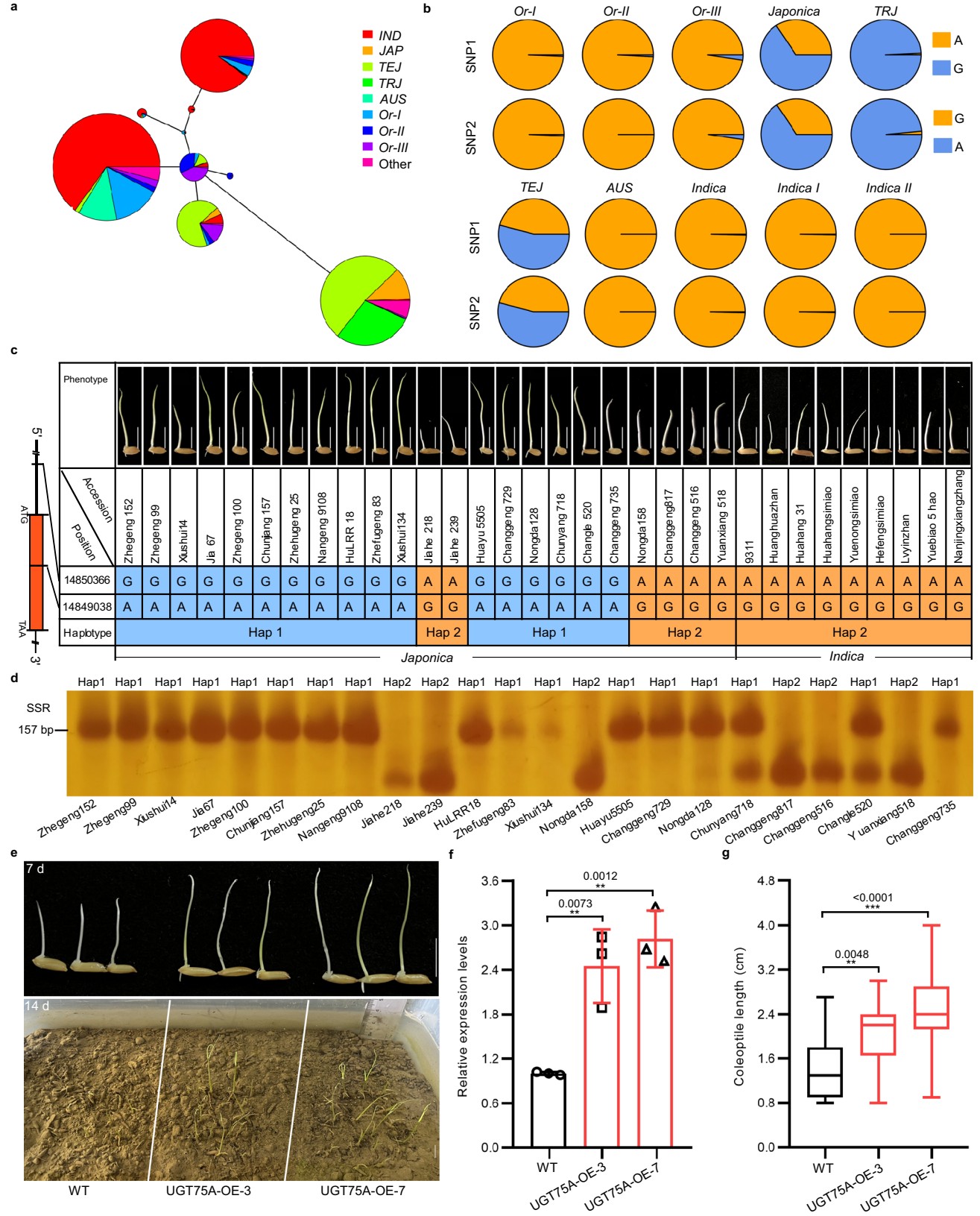

## Methods

### Plant materials

We selected a total of 245 rice accessions from the Rice Diversity Panel 1 (https://www.ars.usda.gov/) (Supplementary Data 1) and collected *japonica* varieties from Jilin Province in northern China (40°52′–46°18′ N, 121°38′–131°9′E) and Zhejiang Province in southern China (27°02′–31°11′N, 118°01′–123°10′ E), as well as *indica* varieties from Guangdong Province in southern China (20°13′–25°31′N, 109°39′–117°19′E) for phenotypic evaluation. In addition, using the CRISPR/Cas9 system, we generated three *OsUGT75A*, *OsJAZ6/7*, and *OsABI3/5* mutants in the *japonica* Nipponbare or Zhonghua11 backgrounds, and used an *Agrobacterium tumefaciens*-mediated

**Fig. 7 | Analysis of *OsUGT75A* haplotypes associated with coleoptile length under submergence. a** An *OsUGT75A* haplotype network. The size of circles is proportional to the number of samples for a given haplotype. *IND indica, JAP japonica, TEJ* temperate *japonica, TRJ* tropical *japonica, Or Oryza rufipogon.* **b** The diversity of elite haplotype 1 (Hap1) in rice accessions. **c** A comparison of elite Hap1 distribution and coleoptile lengths under submergence in the currently cultivated *japonica* and *indica* in China. **d** The development of a simple sequence repeat marker for *OsUGT75A. n = 3* independent experiments. **e** Representative images of coleoptile length under submergence (8 cm depth of water) for 7 days and seedling establishment of direct seeding in soils under submergence (4 cm depth of water) for 14 days among *indica* Huanghuazhan wild-type (WT) and overexpressed

*OsUGT75A* lines (OE-3 and OE-7). Scale bars represent 10 mm. **f** Comparison of the relative expression levels of *OsUGT75A* in WT and overexpressed lines determined by qRT-PCR analysis. Data were presented as mean ± SD, *n = 3* biologically independent samples. Expression is shown relative to that in the WT, the value of which was set to 1, with the *OsActin* gene being used as an internal control. **g** Comparison of the coleoptile lengths of WT and overexpressed lines, *n = 20.* Center lines show the medians, box limits indicate the 25th and 75th percentiles, whiskers extend to the minimal and maximal values as determined by GraphPad Prism 8.0.1 software. In **f, g** significant differences were determined by two-tailed Student's *t* tests (**$P < 0.01$, ***$P < 0.001$). Source data are provided as a Source Data file.

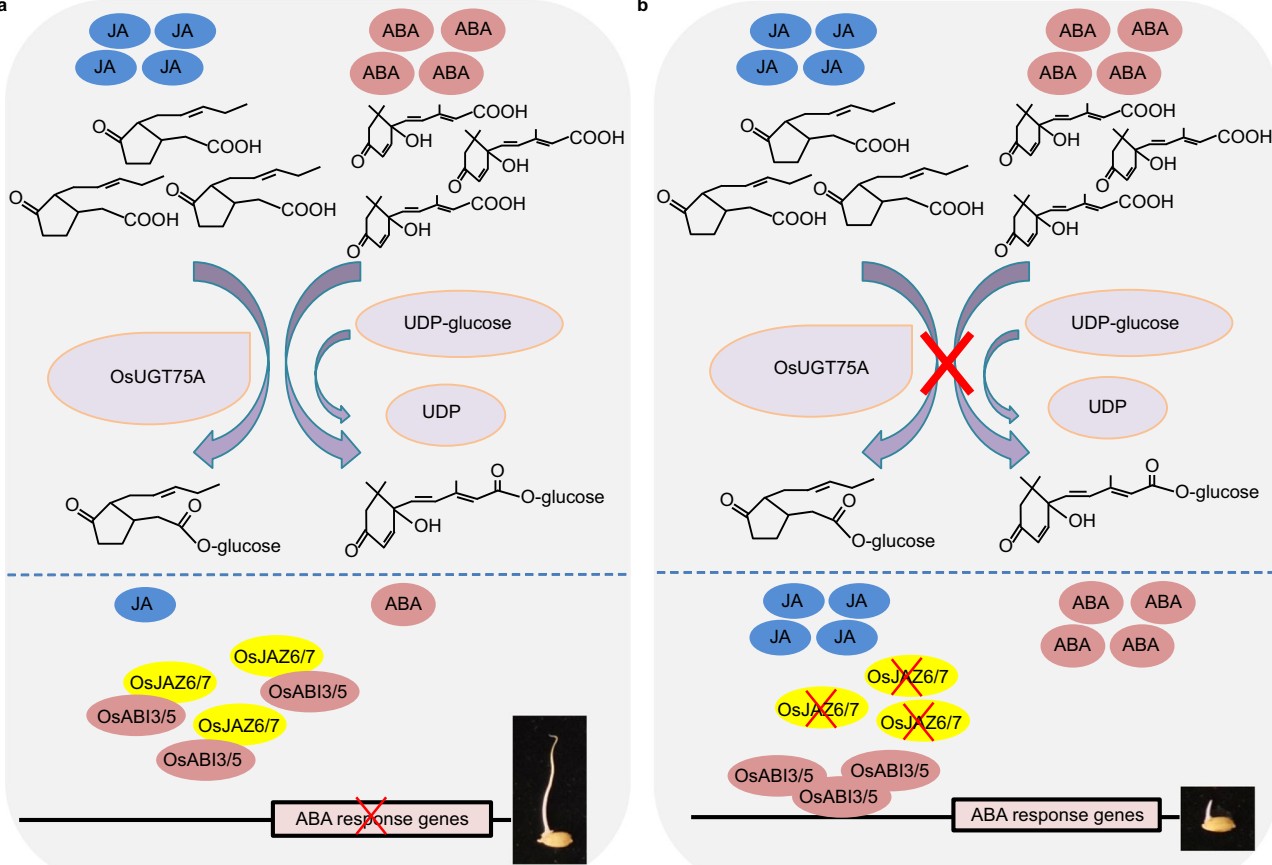

**Fig. 8 | Proposed working model for the role of OsUGT75A in the regulation of rice seed germination under submergence. a** In wild-type plants, OsUGT75A catalyzes the glycosylation of abscisic acid (ABA) and jasmonic acid (JA) contributes to the decrease of free ABA and JA accumulation, which leads to the OsJAZ6/7 proteins interact directly with OsABI3/5 and repress the transcriptional activity of OsABI3/5 and subsequent regulate ABA signaling pathway. **b** In *Osugt75a* mutant

plants, the lack of glucosyltransferase activity results in the accumulation of free ABA and JA, which leads to repressing the interaction between OsJAZ6/7 and OsABI3/5 proteins and activates the transcriptional activity of OsABI3/5 and subsequent regulate ABA signaling pathway. OsUGT75A thereby regulates seed germination under submergence involving the crosstalk of JA and ABA signaling pathways.

transformation system to generate two *OsUGT75A* over-expression lines driven by the *Ubi* promoter in *japonica* Nipponbare background (OE-1 and OE-2) and *indica* Huanghuazhan background (OE-3 and OE-7). The sequences of all primers used for amplification are listed in Supplementary Data 5. Seeds of each material were harvested at the maturity stage and dried at 42 °C for 7 days to break seed dormancy[39]. Well-filled seeds were selected for experiments.

### Genome-wide association analysis
For each of the assessed genotypes, we cultivated 15 dried seeds per replicate in test tubes containing water to a depth of 8 cm at 25 °C under a 12/12-h light/dark photoperiod for 7 days, and thereafter

determined the maximum length of the emerged coleoptiles. In each case, the experiment was repeated three times. Genome-wide association analysis (GWAS) was conducted based on a linear mixed-model in EMMAX using coleoptile length and single-nucleotide polymorphism (SNP) markers with missing variation <0.25 and a minor allele frequency (MAF) > 0.05[29] from a set of 700,000 SNP markers[30]. Among these, we obtained 400,272 SNPs across the entire rice genome, 225,730 and 350,586 SNPs across the *japonica* and *indica* population genome, respectively, with respective significant *P*-value thresholds of 2.50E-08, 4.43E-08, and 2.85E-08[40]. Only those associations for which *P*-values exceeded these thresholds and showed a clear peak-like signals (≥3 significant SNPs) within the 200-kb region around the leading

SNP were considered as single association loci[41,42]. The candidate genes for each association locus were predicted within the 200-kb region around the leading SNP. The corresponding haplotypes were determined using Rice Diversity database (http://www.ricediversity.org/)[30] and ECOGEMS software (https://venyao.xyz/ECOGEMS/)[34].

### Phylogenetic reconstruction of UGT

The amino acid sequences of *Arabidopsis* UGTs were downloaded from NCBI databases (https://www.ncbi.nlm.nih.gov/), and multiple sequence alignments were performed using ClustalW. A phylogenetic tree based on these sequences was constructed using the neighbor-joining method with bootstrap sampling (1000 replicates) using MEGA software version 6.06.

### Gene expression analysis

Total RNA was extracted using an HP Plant RNA Kit (Omega, Atlanta, GA, USA) and first-strand cDNA was synthesized using HiScript Reverse Transcriptase (Vazyme, Nanjing, China) according to the manufacturer's instructions. Quantitative RT-PCR was performed using a CFX96 Real-Time System (Bio-Rad, CA, USA) according to Zhao et al.[43]. The PCR conditions were as follows: 95 °C for 2 min, followed by 40 cycles of 95 °C for 5 s and 60 °C for 10 s. The rice *OsActin* gene was used as an internal control, and the comparative $2^{-\Delta\Delta CT}$ method was used to normalize transcript levels[44]. Three biological replications were performed. For the purpose of GUS staining, we used transgenic plants carrying the *OsUGT75A* promoter-GUS fusion construct in the *japonica* Nipponbare background, with staining patterns being recorded using a digital camera (Nikon Corporation, Tokyo, Japan).

### Expression and purification of a GST-tag protein

The coding sequences of *OsUGT75A* were amplified from the reverse-transcribed RNA isolated in rice varieties Nipponbare seedlings, and these were cloned into pGEX6p plasmids as glutathione-*S*-transferase (GST)-fusion proteins (GST-OsUGT75A). The sequences of the primers used are listed in Supplementary Data 5. The GST-OsUGT75A fusion protein was subsequently expressed in *Escherichia coli* BL21 (DE3) cells and grown in LB medium at 37 °C until reaching an OD$_{600}$ of 0.5. The GST-OsUGT75A protein was induced using 0.3 mM isopropyl-1-thio-β-ᴅ-galactopyranoside (IPTG) and cultures were incubated with constant shaking at 15 °C for 24 h. Protein expression was assessed by sodium dodecyl sulfate (SDS)-polyacrylamide gel electrophoresis (PAGE) analysis. Proteins were purified using a GST•Bind™ Purification Kit (Novagen, Darmstadt, Germany) and quantified based on gel densitometry using a bovine serum albumin protein standard (BSA; Sigma-Aldrich, Milan, Italy).

### HPLC and LC-MS analysis

The GST-OsUGT75A fusion protein was quantified using bovine serum albumin protein standard (BSA). Then, the recombinant OsUGT75A proteins (15-20 μg) were incubated at 37 °C with 10 mM DTT, 100 mM Tris-HCl (pH 7.5), 0.1 mM ABA or 1 mM JA substrates, and 2.5 mM UDP-glucose in a final volume of 200 μL. Reactions were stopped by adding methanol after a 2-h incubation and then centrifuged at 18,800×*g* for 10 min. High-performance liquid chromatography (HPLC) analysis was performed using a Shimadzu LC-16 system (Shimadzu, Japan) equipped with a Wondasil C18-WR column (5 μm × 4.6 × 150 mm). Samples were eluted with solvent A (water containing 0.1% formic acid) and solvent B (acetonitrile containing 0.1% formic acid) under different conditions. Conditions of gradient program for ABA and IAA samples, started at 20% solvent B (0–1 min), increased to 80% solvent B (1–5 min), maintained 80% solvent B (5–10 min), finally ramped back to 20% solvent B (10–15 min); flow rate, 1.0 mL/min; temperature, 40 °C; injection volume: 10 μL. Conditions of gradient program for JA samples, started at 15% solvent B (0–3 min), increased to 90% solvent B (3–13 min), maintained 90% solvent B (13–17 min), finally ramped back to 15% solvent B (17–20 min); flow rate, 0.8 mL/min; temperature, 37 °C; injection volume: 10 μL.

The products were further confirmed by the LC-MS system (Agilent UPLC1290-6540UHD Q-TOF, ASMS, California, USA). The mobile phases of the HPLC system comprised solvent A (methanol) and solvent B (0.2% formic acid in water). The sample (10 μL) was injected onto an Agilent C18 column (50 × 2.1 mm, 1.8 μm) and eluted with solvent A: solvent B = 80%: 20% at a flow rate of 0.3 mL/min for 10 min. The mass spectrometer operated in a positive electro spray ionization mode with a probe voltage of 4.0 kV. Gas temperature and SheathGas temperature were set at 300 °C and 350 °C, respectively. The data acquisition and analysis were performed with MASS HUNTER.

### Hormone quantification

The quantification of endogenous hormones was carried out by Wuhan Metware Biotechnology Co., Ltd, Wuhan, China. Approximately 50 mg fresh weight of each sample was frozen in liquid nitrogen and ground into a fine powder. The sample extracts were analyzed using an LC-ESI-MS/MS system (HPLC: Shim-pack UFLC SHI-MADZU CBM30A system; Shimadzu MS, Applied Biosystems 6500 Triple Quadrupole). The analytical conditions of HPLC were as follows, LC: column, Waters ACQUITY UPLC HSS T3 C18 (100 mm × 2.1 mm, 1.8 μm); solvent system, water with 0.04% acetic acid (A), acetonitrile with 0.04% acetic acid (B); gradient program, started at 5% B (0–1 min), increased to 95% B (1–8 min), 95% B (8–9 min), finally ramped back to 5% B (9.1–12 min); flow rate, 0.35 mL/min; temperature, 40 °C; injection volume: 2 μL. The ESI source operation parameters were as follows: ion source, ESI + /-; source temperature 550 °C; ion spray voltage (IS) 5.5 kV (positive), −4.5 kV (negative); curtain gas (CUR) was set at 35 psi, respectively. Hormone contents were determined using an internal standard method, with three biological replications being performed for each hormone.

### Yeast two-hybrid assays

The coding sequences of *OsJAZ6/7* and *OsABI3/5* were cloned into the pGBKT7 and pGADT7 vectors, respectively, serving as bait and prey. The bait and prey vectors were co-transformed into the yeast strain Y2H Gold, which were plated on SD/Leu-Trp-medium and incubated at 30 °C for 2 days, and the resulting positive clones were cultivated on fresh SD-Trp/-His/-Leu/-Ade plates at 30 °C for 2–4 days. The assay of α-galactosidase activity was conducted using X-α-Gal as a substrate according to a protocol (Coolaber, CAS:107021-38-5, China). The sequences of primers used for vector construction are listed in Supplementary Data 5.

### Bimolecular fluorescence complementation

The coding sequences of *OsJAZ6/7* were cloned into the p2YN vector, yielding OsJAZ6::p2YN and OsJAZ7::p2YN constructs, whereas those of *OsABI3/5* were cloned into the p2YC vector to generate OsABI3::p2YC and OsABI5::p2YC. The recombinant plasmids were introduced into *Agrobacterium* strain GV3101, and overnight agrobacterial cultures were resuspended in infiltration buffer (10 mM MgCl$_2$, 0.1 mM acetosyringone, and 10 mM MES). Subsequently, the culture suspensions were agroinfiltrated into *Nicotiana benthamiana* leaves using a needleless syringe. YFP fluorescence was observed by confocal microscopy (LSM510; Carl Zeiss, Germany). The sequences of primers used for vector construction are listed in Supplementary Data 5.

### Pull-down assay

The coding sequences of *OsABI3/5* were cloned into pGEX6p plasmids as glutathione-*S*-transferase (GST)-fusion proteins (GST::OsABI3 and GST::OsABI5), whereas those of *OsJAZ6/7* were subcloned into the maltose-binding protein (MBP) tag-containing plasmid pMALc2x to generate MBP::OsJAZ6 and MBP::OsJAZ7 fusion proteins. These fusion proteins were expressed in *E. coli* BL21 cells (TSV-A09; Tsingke, China)

and purified in vitro using a TNT quick-coupled transcription/translation system (Promega, Madison, WI, USA) according to manufacturer's instructions. For pull-down assays, 2 µg of MBP::OsJAZ6 or MBP::OsJAZ7 prey proteins was incubated with 2 µg of immobilized GST, GST::OsABI3, or GST::OsABI5 bait proteins. Precipitates were washed three times and visualized on SDS-PAGE gels. Then the pulled down proteins were further analyzed by immunoblotting using anti-GST (Cell Signaling Technology, Danvers, MA, USA, 2624 S, 1:2000 dilution) and anti-MBP (Cell Signaling Technology, Danvers, MA, USA, 2396 S, 1:2000 dilution). The sequences of the primers used are listed in Supplementary Data 5.

## Transient transactivation assay

Sequences of the *OsEM1* promoter (800 bp) were amplified from rice seedlings, and cloned into the pGreenII 0800-LUC vector as reporters. The coding sequences of *OsJAZ6/7* and *OsABI3/5* were amplified and cloned into the pGREENII62-SK vector as effectors. Recombinant plasmids were separately transformed into rice protoplasts. The activities of firefly and *Renilla* (REN) luciferases (LUC) were determined using a dual-luciferase reporter assay system (Promega, Madison, WI, USA) after 12 h to 16 h of incubation at 28 °C in the dark, with relative luciferase activity being calculated as the ratio of LUC to REN (LUC/REN). The sequences of the primers used for vector construction are listed in Supplementary Data 5.

## Data analysis

Experimental data were analyzed using SAS software version 9.4 (Cary, NC, USA). Significant differences among samples were compared using Student's *t*-test or an analysis of variance (ANOVA).

## Reporting summary

Further information on research design is available in the Nature Portfolio Reporting Summary linked to this article.

## Data availability

The amino acid sequences of UGTs were downloaded from NCBI databases [https://www.ncbi.nlm.nih.gov/]. The GWAS genotype data are available on Rice Diversity Project [http://www.ricediversity.org/data/index.cfm]. Source data are provided with this paper.

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

## Acknowledgements

The authors thank USDA Dale Bumpers National Rice Research Center, Stuttgart, Arkansas, for providing rice seeds. This work was supported by the National Natural Science Foundation of China (Grant No. 32201838 to J.Z.; Grant No. 32272157 to Y.H.; Grant No. 32172052 to Z.W.; Grant No. 31971995 to Z.W.), the Natural Science Foundation of Guangdong Province (Grant No. 2023A1515012052 to J.Z.), and the Double First-class Discipline Promotion Project (Grant No. 2021B10564001 to Y.H. and J.Z.).

## Author contributions

Z.W. designed the research; Y.H. and S.S. performed all the major experiments; Z.H., C.H., Z.T., and Q.H. participated in the material preparation; J.Z. and L.P. performed the phenotype evaluation; Z.W., Y.H., and S.S. analyzed the data and wrote the paper. All of the authors discussed the results and commented on the manuscript.

## Competing interests

The authors declare no competing interests.
