## [Peer Review File · Nature Communications]

UDP-glucosyltransferase OsUGT75A promotes submergence tolerance during rice seed germinationReviewers' Comments:

Reviewer #1:

Remarks to the Author:

Rice coleoptile elongation has been a well-documented plant response in which multiple plant hormones are involved. In particular, ethylene and ABA are important regulators in this process in multiple plants, including lumex. The ABA levels decrease when plants are submerged., which is a prerequisite for coleoptile elongation together with increasing GA sensitivity. ABA 8'-hydroxylase is known to be an important enzyme to decrease the ABA levels in submerged plants. Identification of OsUGT75A in rice submergence response is a potentially interesting finding. The followings are my comments and suggestions.

1. Slow induction of OsUGT75A in Fig 2f

This work is preliminary in terms of mechanisms of submergence response.

The ABA decrease in submerged plants is usually a fast response. Submergence-induced ABA 8'-hydroxylase gene is usually fast, such as within 1 hour, coincided with this decrease in plants including rice. Submergence induction is triggered by submergence itself or oxygen deficiency. I would think that kinetics of gene expression may hit well to a hypothesis that induction of OsUGT75A is associated with oxygen deficiency. Alternatively, Fig 2g shows that expression of OsUGT75A is coleoptile-specific, so slow induction of OsUGT75A may be due to the restricted expression in coleoptile.

2. Related to 1, slow induction of OsUGT75A may suggest that the role of UGT75A is to maintain the low ABA levels in submerged plants. I suggest the author to have a timecourse hormone analysis. Timecourse expression of ABA 8'-hydroxylase gene might also strength their argument.

3. Hormone analysis

I suggest the authors to measure hormone levels in coleoptile separated from seeds. For JA, I suggest the authors to analyze JA-Ile, the bioactive form as well.

Overall, finding of OsUGT75A as a major QTL for coleoptile elongation is an interesting finding. On the other hand, interactions between JAZ and ABI3/5 were reported in other species. This downstream event may be worth elucidating, but is not a novel mechanism to be presented here. I believe in depth characterization of submergence response might increase the strength of this work.

Reviewer #2:

Remarks to the Author:

Review He et al. 2022

This manuscript presents the identification of a GWAS QTL on chromosome 11, controlling coleoptile length of germinating rice seeds under submergence conditions. Its causal gene, UDP-glucosyltransferase OsUGT75A, boosts anaerobic germination via decreasing free ABA and JA levels by promoting glycosylation of the two hormones in germinating seeds under the submergence stress. Further, the authors reported that OsUGT75A positive allele also accelerates coleoptile elongation via mediating interactions between OsJAZ and OsABI proteins. The authors also claimed that OsUGT75A can be targeted for breeding via marker-assisted selection or transgenic methods.

Overall, this is an excellent study, revealing the role of OsUGT75A and identifying and confirming its mediating role in interactions between OsJAZ and OsABI proteins. Several complementary experiments have been performed to confirm these findings. Furthermore, the authors have also uncovered the domestication of this gene. Ultimately, OsUGT75A can be applied to improve rice tolerance to submergence during germination, especially for indica rice and other cultivated rice varieties that do not possess the tolerant alleles.

I don't have any major concerns with this manuscript. But the following are my minor comments to improve the clarification of the manuscript:

- Line 112: "Figure 1a" change to "Figure 1a,b"
- Line 396: "drived" change to "driven"
- Line 730: delete "in"
- Line 784: "regulation seed" change to "regulation of seed"
- Line 791: "repress" change to "repressing"
- Line 792: "activate" change to "activates"
- Line 794: delete "in"
- Supplementary Figure 6: It seems to me that some of the variances in the measurement of several hormones are pretty broad. It makes me wonder about the robustness of the statistical analysis. However, further confirmation with exogenous application of the hormones, especially ETH (Suppl. Fig. 7), seemed to confirm the results. But I am a bit confused with the results of GA confirmation since I expect that the application of GA will rescue the coleoptile elongation at some level (?). It will be good to see the confirmation for the auxin as well. But it seemed that it had been confirmed using HPLC (Suppl. Fig. 8).
- Supplementary Table 2 was not cited in the texts.

Reviewer #3:

Remarks to the Author:

He et al. provide a thorough study of an important issue of rice cultivation addressing the agronomically beneficial practice of flooding after direct seeding. Thereby, rice cultivars with a high(er) tolerance to submergence are preferred. This trait is conferred by a more rapid coleoptile growth to reach the water surface allowing a better and earlier supply with oxygen. The authors approach this topic with a classical GWAS study identifying a defined genomic region. With this result they start a really broad and in depth study encompassing not only the identification of the responsible gene (a small-molecule glycosyltransferase, UGT75A), but also quite convincing information on its substrates (ABA and JA) and the physiological role and mechanistic details of UGT75A in the fine-tuning (and suppression of) ABA and JA signals, which eventually leads to enhanced coleoptile growth. They also elude on the breeding history of active and less active UGT75A variants in japonica and indica rice varieties. These studies come along a broad experimental portfolio encompassing GWAS, CRISPR/Cas9 genome modification (generation of loss-of-function UGT and ABA/JA-related signaling molecules), determination of hormone levels, recombinant UGT and LCMS studies, and protein-protein interaction experiments.

However, some of these aspects should be more carefully interpreted:

1) I agree that there are several data supporting the finding that UGT75A catalyzes the glucosylation of JA and ABA. However, recombinant UGT75A produces really minor amounts of the putative glucosides in vitro, in particular JA-glucoside is a really minor product peak (also the obviously correct fragmentation shown by the LCMS study present very weak peak intensities). The fact should at least be mentioned and not left to a more or less experienced reader.

2) A more severe, related problem is the comparison of recombinant UGT75A variants of the active Hap1 and - as claimed by the authors - inactive Hap2 isoform. Since the the Hap1 enzyme has such a low activity, the failure in case of Hap2 may also be related to the recombinant production, not each and every enzyme preparation is optimal. Apart from such a general variability of the assay, there is further concern in this direction, since the Hap2 prep was of much lower quality than the Hap1 prep showing lots of degradation or other impurities (Suppl. Fig. 9).

I suggest to eliminate the claim regarding differential activity. Nevertheless, a presentation of the polymorphism should be presented for both the promoter region and the coding region. The author do have these sequences and they may be more instructive than a single SNP in either the promoter region or the coding sequence. Regarding the promoter, one could demonstrate with minor effort that the lower activity is due to altered binding motifs; regarding the coding region, any altered amino acids will support (or not) that the activity could be changed, since detailed UGT structure and

knowledge about substrate binding sites are available.

3) Does the beneficial UGT75A trait regarding submergence tolerance have any other (negative) effect during later stages of development and growth?

4) Discussion: Line 342, line 374: the authors argue about OsABI proteins activating transcription; however, ABI, i.e. PP2C phosphatases are negative regulators inhibiting SnRK(2)-like proteins at least in Arabidopsis, aren't they? This would of course also require a modification of the model presented in the text and Figure 8.

5) Minor issues

a) Line 190: ABA and JA were enhanced in Osugt75a plants, not reduced (maybe the same mistake was written in another instance?)

b) Line 217 (and a few other cases): typo GUT instead of UGT

c) Line 332: cancel "is"

Reviewer #1 (Remarks to the Author):

Rice coleoptile elongation has been a well-documented plant response in which multiple plant hormones are involved. In particular, ethylene and ABA are important regulators in this process in multiple plants, including *lumex*. The ABA levels decrease when plants are submerged., which is a prerequisite for coleoptile elongation together with increasing GA sensitivity. ABA 8'-hydroxylase is known to be an important enzyme to decrease the ABA levels in submerged plants. Identification of OsUGT75A in rice submergence response is a potentially interesting finding. The followings are my comments and suggestions.

1. Slow induction of OsUGT75A in Fig 2f

This work is preliminary in terms of mechanisms of submergence response. The ABA decrease in submerged plants is usually a fast response. Submergence-induced ABA 8'-hydroxylase gene is usually fast, such as within 1 hour, coincided with this decrease in plants including rice. Submergence induction is triggered by submergence itself or oxygen deficiency. I would think that kinetics of gene expression may fit well to a hypothesis that induction of OsUGT75A is associated with oxygen deficiency. Alternatively, Fig 2g shows that expression of OsUGT75A is coleoptile-specific, so slow induction of OsUGT75A may be due to the restricted expression in coleoptile.

Reply: Yes, we agree with the reviewer that the slow induction of *OsUGT75A* might be associated with oxygen deficiency. We observed that the expression of *OsUGT75A* is gradually increased especially in the late (48 to 72 h) stage of seed germination and in the developing coleoptiles under submergence. This statement has been added in Results section of our revised manuscript.

2. Related to 1, slow induction of OsUGT75A may suggest that the role of UGT75A is to maintain the low ABA levels in submerged plants. I suggest the author to have a timecourse hormone analysis. Timecourse expression of ABA 8'-hydroxylase gene might also strengthen their argument.

Reply: According to the reviewer's suggestion, we added the analysis of the dynamics of ABA and JA levels in the germinating seeds and in the developing coleoptiles under submergence. The new results showed that the levels of free ABA, JA, and JA-Ile were significantly higher in the germinating seeds and in the developing coleoptiles in *Osugt75a* mutants compared to those of WT under submergence. The data have been added in the newly revised Figure 3.

Meanwhile, the expression of ABA 8'-hydroxylase genes was also tested in germinating seeds under submergence. We observed that the expressions of *OsABA8ox* genes, especially *OsABA8ox1* in the early (0 to 12 h) stage of seed germination and *OsABA8ox3* in the middle (12 to 48 h) stage of seed germination, were significantly decreased in *Osugt75a* mutants compared to those of WT seeds under submergence. The data have been added in the revised Supplementary Figure 10.

Taken together, these data suggest that *OsABA8ox* genes might be also involved in the *OsUGT75A*-mediated degradation of ABA under submergence. As mentioned above, the expression of *OsUGT75A* is gradually increased especially in the late (48 to 72 h) stage of seed germination and in the developing coleoptiles under submergence. These results imply that the slow induction of *OsUGT75A* may play a role in maintaining the low levels of free ABA, JA, and JA-Ile in the germinating seeds and in the developing coleoptiles under submergence. These discussions have been added in Results in our revised manuscript.

3. Hormone analysis

I suggest the authors to measure hormone levels in coleoptile separated from seeds.

For JA, I suggest the authors to analyze JA-Ile, the bioactive form as well.

Reply: According to the reviewer's suggestion, the free ABA, JA and JA-Ile levels in the coleoptile have been measured and the newly generated results have been added in the revised Figure 3. These data are consistent with previous conclusion in germinating seeds that the lower free ABA, JA, and JA-Ile were observed in the developing coleoptiles in *Osugt75a* mutants compared to those of WT under submergence.

Overall, finding of *OsUGT75A* as a major QTL for coleoptile elongation is an interesting finding.

On the other hand, interactions between JAZ and ABI3/5 were reported in other species. This downstream event may be worth elucidating, but is not a novel mechanism to be presented here. I believe in depth characterization of submergence response might increase the strength of this work.

Reply: Thanks for the reviewer's suggestion. We agree that elucidating the downstream mechanisms underlying the submergence response regulated by *OsUGT75A* is out of the scope of the current study. We hope we can report the results to the readers in the near future.

Reviewer #2 (Remarks to the Author):

Review He et al. 2022

This manuscript presents the identification of a GWAS QTL on chromosome 11, controlling coleoptile length of germinating rice seeds under submergence conditions. Its causal gene, UDP-glucosyltransferase *OsUGT75A*, boosts anaerobic germination via decreasing free ABA and JA levels by promoting glycosylation of the two hormones in germinating seeds under the submergence stress. Further, the authors reported that *OsUGT75A* positive allele also accelerates coleoptile elongation via mediating interactions between *OsJAZ* and *OsABI* proteins. The authors also claimed that *OsUGT75A* can be targeted for breeding via marker-assisted selection or transgenic methods.

Overall, this is an excellent study, revealing the role of *OsUGT75A* and identifying

and confirming its mediating role in interactions between OsJAZ and OsABI proteins. Several complementary experiments have been performed to confirm these findings. Furthermore, the authors have also uncovered the domestication of this gene. Ultimately, OsUGT75A can be applied to improve rice tolerance to submergence during germination, especially for indica rice and other cultivated rice varieties that do not possess the tolerant alleles.

I don't have any major concerns with this manuscript. But the following are my minor comments to improve the clarification of the manuscript:

- Line 112: "Figure 1a" change to "Figure 1a,b"
- Line 396: "drived" change to "driven"
- Line 730: delete "in"
- Line 784: "regulation seed" change to "regulation of seed"
- Line 791: "repress" change to "repressing"
- Line 792: "activate" change to "activates"
- Line 794: delete "in"

Reply: Thanks for the reviewer's careful language editing. All suggestions have been incorporated in the revised manuscript.

- Supplementary Figure 6: It seems to me that some of the variances in the measurement of several hormones are pretty broad. It makes me wonder about the robustness of the statistical analysis. However, further confirmation with exogenous application of the hormones, especially ETH (Suppl. Fig. 7), seemed to confirm the results. But I am a bit confused with the results of GA confirmation since I expect that the application of GA will rescue the coleoptile elongation at some level (?). It will be good to see the confirmation for the auxin as well. But it seemed that it had been confirmed using HPLC (Suppl. Fig. 8).

Reply: Thanks for the reviewer's comments. In regard to variances of hormones measurement, the amount of cytokinin is relatively lower than the other hormones. The variations of cytokinin levels are larger than that would be expected for hormones accumulated at higher levels. Indeed the variations of the other hormones are within normal ranges.

Our exogenous applications of the ETH showed that the ETH didn't improve the coleoptile length of *Osugt75a* mutants (Supplementary Figure 7). Thus, we speculated that *OsUGT75A* might regulate the coleoptile length independent of ETH.

For GA function on the coleoptile elongation, we observed that 5 μ M GA3 treatment can increase the coleoptile elongation of WT but not in *Osugt75a* mutants (Supplementary Figure 7). Thus, we speculated that *OsUGT75A* might regulate the coleoptile length independent of GA pathway.

The reviewer is correct. We did HPLC analysis of IAA in the previous version of this manuscript. The results showed that *OsUGT75A* does not influence auxin level under submergence. According to the reviewer's suggestion, we added the IAA treatment experiment. However, we didn't observe any distinguishable responses to IAA in *Osugt75a* mutants. The new data have been added in Supplementary Figure 8 of the revised manuscript.

- Supplementary Table 2 was not cited in the texts.

Reply: Thanks for the reviewer's careful reading. We added the citation of Supplementary Table 2 in the revised text.

Reviewer #3 (Remarks to the Author):

He et al. provide a thorough study of an important issue of rice cultivation addressing the agronomically beneficial practice of flooding after direct seeding. Thereby, rice cultivars with a high(er) tolerance to submergence are preferred. This trait is conferred by a more rapid coleoptile growth to reach the water surface allowing a better and earlier supply with oxygen. The authors approach this topic with a classical GWAS study identifying a defined genomic region. With this result they start a really broad and in depth study encompassing not only the identification of the responsible gene (a small-molecule glycosyltransferase, UGT75A), but also quite convincing information on its substrates (ABA and JA) and the physiological role and mechanistic details of UGT75A in the fine-tuning (and suppression of) ABA and JA signals, which eventually leads to enhanced coleoptile growth. They also elude on the breeding history of active and less active UGT75A variants in japonica and indica rice varieties. These studies come along a broad experimental portfolio encompassing GWAS, CRISPR/Cas9 genome modification (generation of loss-of-function UGT and ABA/JA-related signaling molecules), determination of hormone levels, recombinant UGT and LCMS studies, and protein-protein interaction experiments.

However, some of these aspects should be more carefully interpreted:

1) I agree that there are several data supporting the finding that UGT75A catalyzes the glucosylation of JA and ABA. However, recombinant UGT75A produces really minor amounts of the putative glucosides in vitro, in particular JA-glucoside is a really minor product peak (also the obviously correct fragmentation shown by the LCMS study present very weak peak intensities). The fact should at least be mentioned and not left to a more or less experienced reader.

Reply: Thanks for the reviewer's suggestions. We agree that the activity of OsUGT75A glucosylation of JA is low in this study. The possible reason could be that the reaction system/condition is not optimal. As suggested, we have added some discussion on this point in results of OsUGT75A glycosylation of ABA and JA.

2) A more severe, related problem is the comparison of recombinant UGT75A variants of the active Hap1 and - as claimed by the authors - inactive Hap2 isoform. Since the Hap1 enzyme has such a low activity, the failure in case of Hap2 may also be related to the recombinant production, not each and every enzyme preparation is optimal. Apart from such a general variability of the assay, there is further concern in this direction, since the Hap2 prep was of much lower quality than the Hap1 prep

showing lots of degradation or other impurities (Suppl. Fig. 9).

I suggest to eliminate the claim regarding differential activity. Nevertheless, a presentation of the polymorphism should be presented for both the promoter region and the coding region. The author do have these sequences and they may be more instructive that a single SNP in either the promoter region or the coding sequence. Regarding the promoter, one could demonstrate with minor effort that the lower activity is due to altered binding motifs; regarding the coding region, any altered amino acids will support (or not) that the activity could be changed, since detailed UGT structure and knowledge about substrate binding sites are available.

Reply: We agree with the reviewer that the recombinant production may influence the activity of Hap1 and Hap2 isoform. It will affect the accurate comparison of recombinant UGT75A variants between the active Hap1 and inactive Hap2 isoform. Thus, according to the reviewer's suggestion, we have deleted the data of different activity between Hap1 and Hap2 in our revised manuscript.

3) Does the beneficial UGT75A trait regarding submergence tolerance have any other (negative) effect during later stages of development and growth?

Reply: According to the reviewer's suggestion, we compared agronomic traits among wild-type, *Osugt75a* mutants, and overexpression lines. We observed increased number of secondary branch and number of grains per panicle in overexpression *OsUGT75A* lines compared with WT, whereas plant height, grain length, grain width, and grain weight were decreased in overexpression lines. The data have been added in the new Supplementary Figure 16 in our revised manuscript.

4) Discussion: Line 342, line 374: the authors argue about OsABI proteins activating transcription; however, ABI, i.e. PP2C phosphatases are negative regulators inhibiting SnRK(2)-like proteins at least in Arabidopsis, aren't they? This would of course also require a modification of the model presented in the text and Figure 8.

Reply: We agree with the reviewer on this point. Several Arabidopsis ABIs such as PP2C phosphatases are negative regulators inhibiting SnRK(2)-like proteins. According to the reviewer's suggestion, we have revised the description of "transcriptional activation" as "transcriptional activity" in the text of our revised manuscript. The arrow lines indicating the activating transcription of ABA response genes have been deleted in the Figure 8.

5) Minor issues

a) Line 190: ABA and JA were enhanced in *Osugt75a* plants, not reduced (maybe the same mistake was written in another instance?)

b) Line 217 (and a few other cases): typo GUT instead of UGT

c) Line 332: cancel "is"

Reply: Thanks for the reviewer's careful editing. All these suggestions have been incorporated in the revision.

Reviewers' Comments:

Reviewer #1:

Remarks to the Author:

The authors revised manuscript correctly.

Reviewer #2:

Remarks to the Author:

Thank you for all the revisions!

Reviewer #3:

Remarks to the Author:

Thanks for the additional data and explanations. Many of them are conclusive and helpful. However, I do still have some major problems with the conclusion, in part even enhanced by the explanations and additional data.

1) The abstract specifically addresses the action of UGT75A as "promoting the glycosylation of the two phytohormones", i.e. ABA and JA. However, there is not a single measurement of ABA or JA glucosides! Instead, this notion is based on (i) the increase of free ABA and JA in *ugt75a* loss-of-function mutants and (ii) be the action of the recombinant protein. In fact, (i) could be explained in that manner, but there may be also alternative measures and the effect on the two phytohormones is a downstream consequence of the glycosylation of yet another signaling molecule X. This may be really relevant, since the *in vitro* data claiming these activities are very weak for different reasons (which may be related): first of all, both activities, in particular the activities towards JA (but also ABA) are very weak. Here, I also take into account a two hour incubation with a high amount of enzyme (see also next issue), which should usually give a decent product peak that is clearly lacking. Thus, the reaction conditions may enforce the detection of an artefactual side activity. On the other hand, the enzyme preps are in the range of "lousy to acceptable", but this is not clear at all: Figure 4a and b show an almost fully degraded enzyme prep (only minor amounts of potentially full-length protein are left), whereas 4 b shows a decent prep, still with a considerable amount of degradation. Both figures are shown without any explanation on the reason to show them together!? Regarding the experimental side of the enzyme assays: what does it mean to use 5-10 μ g enzyme per assay: was there a scan of the gel lane and quantitative estimate on the portion of potentially full-length and potentially active protein in the very preps, e.g. probably just some 10-20% in the case of Figure 4a: as a consequence e.g. 5x the amount of total protein, i.e. 25-50 μ g of the prep have been used to account for 5-10 μ g of the enzyme?

2) Impact of GA: in fact, it may be formally correct that UGT75A action does not require GA, however, equally important or even more important is the fact that GA action REQUIRES UGT75A. This should also be worth considering the role of UGT75A.

3) Along these lines, it is also interesting that genes involved in ABA oxidation are repressed in *ugt75a* mutants: this may also argue for or at least allow an alternative mechanism apart from the not convincingly show ABA- and JA glucosylation (I repeat the missing data on ABA and JA glucosides). *ugt75a* mutants may enhance ABA by suppressing ABA oxidation by currently unknown means: this may enhance ABA as well.

Reviewer #3 (Remarks to the Author):

Thanks for the additional data and explanations. Many of them are conclusive and helpful. However, I do still have some major problems with the conclusion, in part even enhanced by the explanations and additional data.

1) The abstract specifically addresses the action of UGT75A as “promoting the glycosylation of the two phytohormones”, i.e. ABA and JA. However, there is not a single measurement of ABA or JA glucosides! Instead, this notion is based on (i) the increase of free ABA and JA in *ugt75a* loss-of-function mutants and (ii) be the action of the recombinant protein. In fact, (i) could be explained in that manner, but there may be also alternative measures and the effect on the two phytohormones is a downstream consequence of the glycosylation of yet another signaling molecule X. This may be really relevant, since the *in vitro* data claiming these activities are very weak for different reasons (which may be related): first of all, both activities, in particular the activities towards JA (but also ABA) are very weak. Here, I also take into account a two hour incubation with a high amount of enzyme (see also next issue), which should usually give a decent product peak that is clearly lacking. Thus, the reaction conditions may enforce the detection of an artefactual side activity. On the other hand, the enzyme preps are in the range of “lousy to acceptable”, but this is not clear at all: Figure 4a and b show an almost fully degraded enzyme prep (only minor amounts of potentially full-length protein are left), whereas 4 b shows a decent prep, still with a considerable amount of degradation. Both figures are shown without any explanation on the reason to show them together!/? Regarding the experimental side of the enzyme assays: what does it mean to use 5-10 µg enzyme per assay: was there a scan of the gel lane and quantitative estimate on the portion of potentially full-length and potentially active protein in the very preps, e.g. probably just some 10-20% in the case of Figure 4a: as a consequence e.g. 5x the amount of total protein, i.e. 25-50 µg of the prep have been used to account for 5-10 µg of the enzyme?

Reply: We agree with the reviewer that the enzyme preparation is the curial step for the detection of enzyme activity. We are sorry that we didn't include enzyme

preparation details in the previous version of the manuscript. In this revised version, we specified that “fusion proteins of OsUGT75A with the GST tag were heterologously expressed in *Escherichia coli* and then quantified for the enzyme activity assay (Figure 4a)” in Results (line 159-160), and “The GST-OsUGT75A fusion protein was quantified using bovine serum albumin protein standard (BSA) by gel densitometry” in the Figure 4a legend. In addition, we specified in Method section (line 420-421) that we increased recombinant protein amount from 5-10 µg to 15-20 µg for the enzyme activity assay. In detail, the concentration of recombinant OsUGT75A protein was estimated by gel densitometry. The results showed that there was approximately 1.5-2 µg in the 20 µL enzyme preparation solution. We used 200 µL enzyme preparation solution, which is equal to 15-20 µg of recombinant OsUGT75A protein, for the enzyme activity assay. Meanwhile, the HPLC conditions were optimized in our revised manuscript. Specifically, conditions of JA gradient program were changed to: started at 15% solvent B (acetonitrile containing 0.1% formic acid) for 3 min, increased to solvent 90% B for 10 min, maintained 90% solvent B 4 min, and finally ramped back to 15% solvent B for 3 min; flow rate was set at 0.8 mL/min; temperature was maintained at 37°C; injection volume was 10 µL. Compared with the previous HPLC conditions, the prolonged time of maintaining 90% solvent B and the shortened time of 15% solvent B were used in the revised HPLC conditions. These details have been revised in Methods (line 429-436). Due to the increased input enzyme amount, and under the optimized HPLC conditions, the signals of ABA-glucose conjugate (ABA-Glu) and putative JA-glucose conjugate (JA-Glu) became much stronger than that in the previous experiments (Figure 4b,c). Overall, our new *in vitro* data suggest that OsUGT75A can catalyze the glycosylation of ABA and JA. These data have been added in Results and in the revised Figure 4 in our revised manuscript.

To further confirm the function of OsUGT75a *in planta*, the endogenous hormones were extracted from the coleoptiles in Nipponbare wild-type (WT) and *Osugt75a*

mutants under submergence (8 cm depth of water) for 7 days. Then, the contents of ABA-Glu and the relative levels of JA-Glu were detected using HPLC (Figure 4 e and f). As expected, significant lower ABA-Glu contents were observed in *Osugt75a* mutants than that in WT (Figure 4 e). Since JA-Glu is not commercially available, we determined the relative levels using targeted HPLC (Supplementary Figure 10). Similar to ABA-Glu, the relatively lower JA-Glu levels were observed in *Osugt75a* mutants compared with that of WT (Figure 4f).

Taken together the newly added/ revised *in vitro* and *in planta* data, we are confident that we have sufficient evidences to support the conclusion that OsUGT75A reduced the level of free ABA and JA by catalyzing the glycosylation ABA and JA during submergence.

2) Impact of GA: in fact, it may be formally correct that UGT75A action does not require GA, however, equally important or even more important is the fact that GA action REQUIRES UGT75A. This should also be worth considering the role of UGT75A.

Reply: We agree with the reviewer that GA regulation of coleoptile growth might require OsUGT75A. We observed that GA₃ treatments can increase the coleoptile elongation of WT but not in *Osugt75a* mutants (Supplementary Figure 7). Further, our qRT-PCR analysis showed that the expression of *OsUGT75A* was significantly induced by GA₃ treatment in WT during seed germination under submergence (Supplementary Figure 9). It suggests that the GA promotion of coleoptile growth might be through increasing *OsUGT75A* expression. These data have been added in Results (line 144-149) and discussions have been added in Discussion section (line 302-303) in our revised manuscript.

3) Along these lines, it is also interesting that genes involved in ABA oxidation are repressed in *ugt75a* mutants: this may also argue for or at least allow an alternative mechanism apart from the not convincingly show ABA- and JA glucosylation (I

repeat the missing data on ABA and JA glucosides). *ugt75a* mutants may enhance ABA by suppressing ABA oxidation by currently unknown means: this may enhance ABA as well.

Reply: We agree with the reviewer that *Osugt75a* mutants may enhance ABA by suppressing ABA oxidation by currently unknown mechanism. Indeed, our data showed that the expressions of *OsABA8ox* genes, especially *OsABA8ox1* in the early stage of seed germination and *OsABA8ox3* in the middle stage of seed germination, were significantly decreased in *Osugt75a* mutants compared to those of WT under submergence. However, we don't know how *OsUGT75A* regulates ABA oxidation, which deserves exploration in our future study. In this revision, we added *OsABA8ox* genes expression data in Supplementary Figure 12. These results were included in the revised manuscript (line 181-184) and discussed in Discussion section (line 297-299).

Revised Figure 4:

Figure 4. Glycosylation ABA and JA by *OsUGT75A*. (a) The *GST-OsUGT75A* fusion protein was quantified using bovine serum albumin protein standard (BSA). *GST-UGT75A* fusion protein catalyzes the glycosylation of ABA (b) and JA (c) *in vitro*, but the *GST* tag could not. Arrow indicates the product ABA-Glu and putative JA-Glu. (d) JA-Glu formed in (c) was confirmed by MS/MS analysis in positive ionization mode. The molecular weight (M) of JA-Glu is 372.27. The reaction

products had the ion peaks at m/z 395.2202 ($M + Na^+$), m/z 210.1180 ($M-H-Glu$), and m/z 166.0911 ($M-COOH-Glu$). Quantification of absolute amounts of ABA-Glu (e) and the relative levels of JA-Glu (f) in coleoptiles in Nipponbare wild-type (WT) and *Osugt75a* mutants under submergence (8 cm depth of water). Absolute amounts of ABA-Glu were measured by HPLC using the commercial available chemical standard. Relative levels of JA-Glu were determined by targeted HPLC. The level of JA-Glu is shown relative to that in WT, the value of which was set to 1. Each column represents the mean \pm standard deviation, $n=3$. * and ** indicate significant differences at the 5% and 1% levels, respectively, as determined by Student's *t*-test between WT and the corresponding mutant.

Newly added Supplementary Figure 9:

Supplementary Figure 9. Expression of *OsUGT75A* regulated by GA₃ treatment in *japonica* Nipponbare under submergence (8 cm depth of water) during seed germination. The relative expression levels of *OsUGT75A* were determined by qRT-PCR analysis. *OsActin* gene was used as an internal control. Each column represents the mean \pm standard deviation, $n=3$; small symbols denote the values of each replicate. ** indicates significant differences at the 1% level as determined by Student's *t*-test between H₂O and the corresponding treatment.

Newly added Supplementary Figure 10:

Supplementary Figure 10. Identification of JA-Glu in coleoptiles in Nipponbare wild-type (WT) and *Osugt75a* mutants under submergence (8 cm depth of water) by targeted HPLC assay. Arrow indicates the putative product JA-Glu.

Reviewers' Comments:

Reviewer #3:

Remarks to the Author:

I thank He et al. for their additional efforts to clarify matters by providing additional data. The revised version is now convincingly showing the impact of UGT75A on the glucosylation of ABA and JA, other issues are discussed more broadly and carefully.

I only have to minor remarks:

- 1) when referring to the densitometric quantification of recombinant GST-OsUGT75A, please add something like full-length or 79 kDa .. to the remark in the legend, i.e. "The full-length GST-OsUGT75A fusion protein ...". This clarifies that you thereby quantify the relevant part of the protein preparation.
- 2) Line 148: "through" instead of "though"

Reviewer #3 (Remarks to the Author):

I thank He et al. for their additional efforts to clarify matters by providing additional data. The revised version is now convincingly showing the impact of UGT75A on the glucosylation of ABA and JA, other issues are discussed more broadly and carefully.

I only have to minor remarks:

1) when referring to the densitometric quantification of recombinant GST-OsUGT75A, please add something like full-length or 79 kDA .. to the remark in the legend, i.e. "The full-length GST-OsUGT75A fusion protein ...". This clarifies that you thereby quantify the relevant part of the protein preparation.

Reply: Yes, "The GST-OsUGT75A fusion protein....." has been changed as "The full-length GST-OsUGT75A fusion protein....." in the legends of Figure 4.

2) Line 148: "through" instead of "though"

Reply: Yes, "though" has been changed as "through".